# Neonatal Wnt-dependent Lgr5 positive stem cells are essential for uterine gland development

Ryo Seishima[1], Carly Leung[1], Swathi Yada[1], Katzrin Bte Ahmed Murad[1], Liang Thing Tan [1], Amin Hajamohideen[1], Si Hui Tan [1], Hideki Itoh[2], Kazuhiro Murakami[3], Yoshihiro Ishida [4], Satoshi Nakamizo[2], Yusuke Yoshikawa[1], Esther Wong[1] & Nick Barker[1,3,5]*

Wnt signaling is critical for directing epithelial gland development within the uterine lining to ensure successful gestation in adults. Wnt-dependent, *Lgr5*-expressing stem/progenitor cells are essential for the development of glandular epithelia in the intestine and stomach, but their existence in the developing reproductive tract has not been investigated. Here, we employ *Lgr5-2A-EGFP/CreERT2/DTR* mouse models to identify *Lgr5*-expressing cells in the developing uterus and to evaluate their stem cell identity and function. *Lgr5* is broadly expressed in the uterine epithelium during embryogenesis, but becomes largely restricted to the tips of developing glands after birth. In-vivo lineage tracing/ablation/organoid culture assays identify these gland-resident Lgr5$^{high}$ cells as Wnt-dependent stem cells responsible for uterine gland development. Adjacent Lgr5$^{neg}$ epithelial cells within the neonatal glands function as essential niche components to support the function of Lgr5$^{high}$ stem cells ex-vivo. These findings constitute a major advance in our understanding of uterine development and lay the foundations for investigating potential contributions of Lgr5$^+$ stem/progenitor cells to uterine disorders.

[1] A*STAR Institute of Medical Biology, Singapore 138648, Singapore. [2] A*STAR Skin Research Institute of Singapore, Singapore 138648, Singapore. [3] Cancer Research Institute, Kanazawa University, Kakuma-machi, Kanazawa 920-1192, Japan. [4] Department of Dermatology, Kyoto University, Yoshida-Konoe-cho, Sakyo-ku 606-8501, Japan. [5] School of Biological Sciences, Nanyang Technological University, Singapore 308232, Singapore. *email: Nicholas.barker@imb. a-star.edu.sg

The majority of the female reproductive tract, comprising the oviduct, uterus, cervix, and upper vagina, develops from the Müllerian duct (Md) during embryogenesis[1]. These organs remain relatively immature at birth and undergo further development during prepuberty to ensure fertility in adulthood. Particularly in uterus, gland development is essential for successful pregnancy as their secretions and products impact on implantation, stromal cell decidualization, and placental development[2].

Uterine glands are essential for proper uterine function as they secrete various factors, such as leukemia inhibitory factor (LIF) that are important for endometrial receptivity and embryo implantation[3]. Their development begins postnatally, involving budding, tubulogenesis, coiling and branching of luminal epithelia, orchestrated by interactions with the underlying stroma[4]. Genetic knockout studies have identified multiple, predominantly Wnt-related, genes required for development of the uterine epithelium[5], but the cellular origins of the glandular epithelium (GE) remain poorly understood. Although single cell sequencing studies have recently documented extensive cellular heterogeneity within the developing mouse uterus epithelium, it is currently unknown whether dedicated endometrial stem/progenitor cells are present[6].

Leucine-rich repeat containing G-protein-coupled receptor-5 (Lgr5) is a facultative component of the Wnt receptor complex that marks Wnt-regulated stem cells responsible for the development, maintenance, and regeneration of multiple epithelia[7–12]. In healthy adult mouse uterus, endogenous Lgr5 expression is low, but is markedly upregulated in response to ovariectomy and downregulated by sex hormone stimulation[13]. In contrast, Lgr5 is robustly expressed in prepubertal mouse endometrium[13]. The identity and function of these endometrial Lgr5+ populations are currently unknown.

Here, we employ non-variegated reporter, in vivo lineage tracing and in vivo ablation mouse models, together with ex vivo organoid culture technologies, to document Lgr5-expressing cells in the developing uterus from embryo to postnatal prepuberty and to evaluate their physiological roles in driving uterine development. We show that Lgr5 is broadly expressed in the Md during embryogenesis, but becomes largely restricted to the tips of developing glands after birth. These region-restricted Lgr5high endometrial cells are Wnt-responsive stem/progenitor cells that are indispensable for uterine gland development. We further identify a distinct hierarchy among developing endometrial cells, with Lgr5high stem/progenitor cells being supported by an epithelial niche that comprises differentiated cells expressing essential Wnt ligands.

## Results

***Lgr5 expression persists through endometrial development.*** To evaluate endogenous Lgr5 expression in female reproductive tracts during development, we examined tissues at various time points during mouse embryogenesis. At embryonic day 12.0 (E12.0), when Md formation is initiated by invagination of the coelomic epithelium[14], nascent Lgr5 expression is observed within the coelomic epithelium, as documented by highly sensitive RNA in situ hybridization (ISH) (Fig. 1a; dashed red line). Of note, Lgr5 was also expressed in the Wolffian duct (Wd) at this time point (Fig. 1a; dashed black line). As a complementary approach, we employed independent *Lgr5-2A-enhanced green fluorescent protein* (EGFP) reporter mice (Fig. 1b). Unlike our previous, variegated *Lgr5*-driven reporter mouse model (*Lgr5-EGFP-ires-CreERT2*[7]), this new model maintains physiological Lgr5 expression levels and consequently faithfully reports all endogenous Lgr5+ populations (Supplementary Fig. 1a). Lgr5-

EGFP expression was detected in coelomic epithelium at E12.0 and colocalized with expression of *Lim1*, a marker of both Md and Wd at this age[15,16] (Fig. 1c). At E12.5, during elongation of the Md, uniform *Lgr5-EGFP* expression was maintained throughout the duct, as well as in Wd (Fig. 1d, e). At postnatal day 0 (P0), robust *Lgr5-EGFP* expression in the uterus was restricted to the epithelium, where it was broadly distributed (Fig. 1f). In contrast, *Lgr5-EGFP* was weakly expressed in oviduct and upper vagina, both of which originate from Md (Supplementary Fig. 1b, c). These data define the origin of Lgr5 expression in the developing female reproductive tract as cells in the nascent Md. At the time of birth, expression is maintained within the epithelial lining of the developing uterus, as well as in oviduct and upper vagina.

**Lgr5+ Müllerian duct cells generate multiple tissues.** We have previously employed in vivo lineage tracing to document the endogenous stem/progenitor cell identity of Lgr5+ populations in a variety of tissues[7,9,11,12]. Here, we adopted the same strategy to evaluate the stem/progenitor cell potential of the embryonic Lgr5+ cells identified within the developing reproductive tract. Lineage tracing was initiated in E11.5 *Lgr5-2A-CreERT2; R26-tdTomato* mice[12] (Fig. 2a) by IP injection of a single 0.2 mg/g body weight dose of Tamoxifen (TAM) to pregnant females. After 24 h, *tdTomato* reporter gene expression was activated in single cells within the Lim1+ Md, consistent with the localization of endogenous Lgr5+ cells at this stage (Fig. 2b). Note that tdTomato (tdTom+)-expressing cells were also evident within the Wd, coinciding with endogenous *Lgr5* expression (Fig. 2b; dashed white line). After 48 h, there was a marked expansion of the Lgr5+ cell-derived tdTom+ tracing units in Lim1+ populations as Md elongation progressed. In contrast, tdTom+ cells were lost from the Wd, likely reflecting the degeneration of this tissue from E13.5 onwards in females (Fig. 2c; dashed white line). At P90, when the uterus had fully matured, contiguous tdTom+ patches of Lgr5+ cell-derived progeny were evident throughout the epithelia of oviduct, uterus, and upper vagina (Fig. 2d). These results identify the Md-resident Lgr5+ cells as being embryonic stem/progenitor cells contributing to the development and maintenance of the epithelia of the female reproductive tract.

***Lgr5 expression is enriched in postnatal endometrial glands.*** We next evaluated Lgr5 expression in female reproductive tracts after birth. *EGFP* expression in *Lgr5-2A-EGFP* reporter mice showed that *Lgr5* expression becomes restricted to the uterus and silenced in oviduct and upper vagina by postnatal day 7 (P7) (Fig. 3e, g, Supplementary Fig. 2e, f, i, j). Thus, we performed detailed *Lgr5* expression analyses on uterus to formally document the localization and identity of the Lgr5+ populations during its development. Q-PCR analyses of wild-type uterus tissues harvested between P3 to early adulthood (P28) revealed that *Lgr5* expression dramatically increased in the first 2 weeks after birth, before gradually declining to low levels at P28 (Fig. 3a). Of note, dioestrus stage in adulthood showed the highest expression during each estrous cycle, implicating a likely hormonal influence on *Lgr5* expression in vivo (Fig. 3b). Detailed *Lgr5* ISH analyses identified endogenous *Lgr5* transcripts distributed at random throughout the developing endometrium at P3, prior to the onset of adenogenesis (Fig. 3d). At P7, which coincides with the onset of gland development[6,17], *Lgr5* expression presented as a gradient, with highest levels present within the nascent gland buds (Fig. 3f). From P14 to P28, this expression gradient became more pronounced, with *Lgr5high* cells predominantly restricted to the tips of developing gland epithelia (GE) (Fig. 3h, j, l). In contrast, *Lgr5* expression was greatly reduced post-puberty, with only the

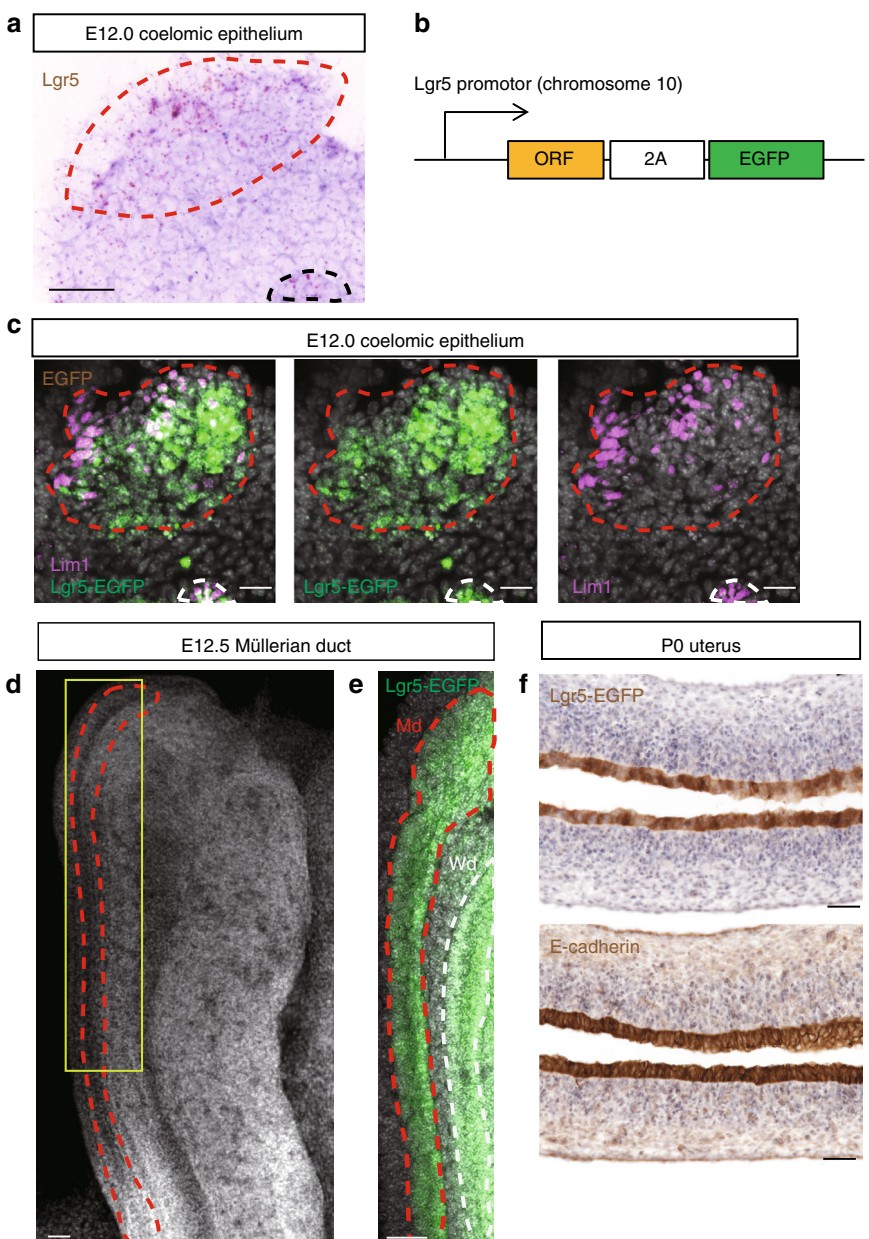

**Fig. 1** *Lgr5* is expressed in the early female reproductive tract during embryogenesis. **a** RNA ISH for *Lgr5* in coelomic epithelium at E12.0. **b** The *Lgr5-2A-EGFP* mouse model employed to evaluate endogenous *Lgr5* expression. **c** Co-IF for Lgr5-EGFP and Lim1 in coelomic epithelium at E12.0. **d** Confocal z-stack image of a whole-mount E12.5 Müllerian duct (highlighted by the red dashed line). Yellow box indicates the region magnified in **e**. **e** Endogenous EGFP fluorescence in E12.5 *Lgr5-2A-EGFP* mouse at Md. **f** Immunostaining for Lgr5-EGFP and E-cadherin in P0 uterus. Dashed red lines indicate Md, and dashed black or white lines indicate Wd, respectively. Md, Müllerian duct; Wd, Wolffian duct; Scale bars, 50 μm. All images are representative of three independent mice.

luminal epithelium (LE) of the dioestrus uterus presenting readily detectable levels of *Lgr5* transcripts (Supplementary Fig. 2a–d). An identical *Lgr5-EGFP* expression profile was documented using *Lgr5-2A-EGFP* reporter mice, further validating this model as a faithful reporter of endogenous *Lgr5* expression in the reproductive tract (Fig. 3h–k). Of note, *Lgr5high* cells were localized to the anti-mesometrial side of the tissue, consistent with a previous report that Wnt signaling is active in this region (Supplementary Fig. 3a, b)[18]. We further employed RNA ISH to document expression of the highly related genes *Lgr4* and *Lgr6*, which are known to modulate Wnt signaling on stem cells in other tissues[19]. In P14 uterus, *Lgr4* expression was readily detectable in the epithelium, but *Lgr6* expression was absent. *Lgr4* expression was

slightly higher in LE than in GE, in contrast to the expression pattern of *Lgr5* (Supplementary Fig. 3c). In the mature uterus, *Lgr4* expression was detected both in LE and GE, whereas *Lgr6* expression was only detected in stroma during dioestrus and prooestrus (Supplementary Fig. 3d).

**Stage-dependent role of Lgr5+ stem cells in endometrium**. To evaluate the stem cell potential of endometrial Lgr5+ cells in prepubertal uterus, we initiated in vivo lineage tracing using 4-hydroxytamoxifen (4OHT) or TAM at a single, low dose that does not impact glandular development or uterine function in *Lgr5-2A-CreERT2*; *R26-tdTomato* mice at P7 or P14

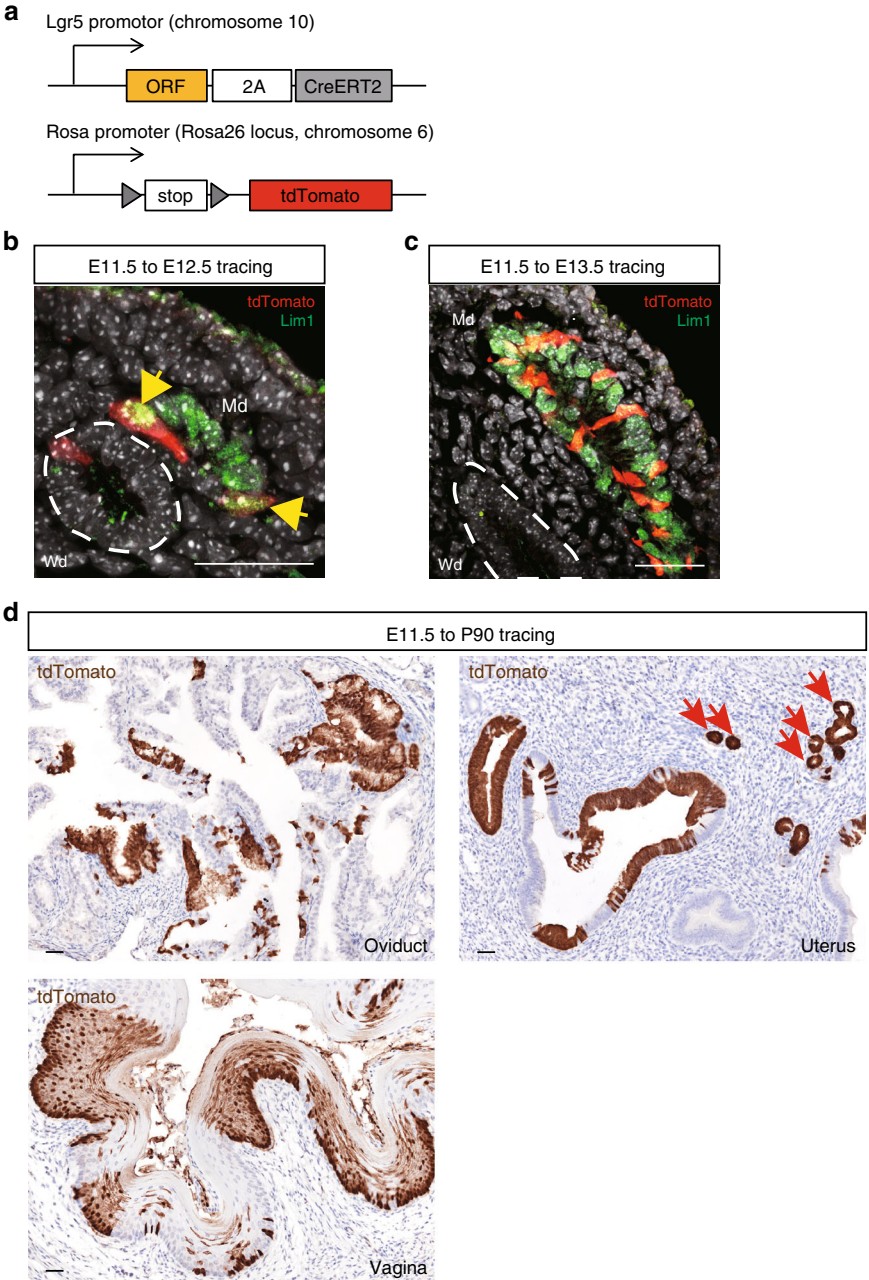

**Fig. 2** Embryonic Lgr5+ populations are stem/progenitor cells for the female reproductive tract. **a** The *Lgr5-2A-CreERT2; R26-tdTomato* mouse model employed to trace *Lgr5+* cell-derived progeny. **b**, **c** Short-term lineage tracing in the developing reproductive tract induced at E11.5. Co-IF for tdTomato and Lim1 on E12.5 genital ducts shows tdTom+ cells in both the Lim1+ Md and the adjacent Lim1− Wd cells (white dashed lines) (**b**) and reveals expansion of Lim1+/tdTom+ tracing exclusively within the Md at E13.5 (**c**). Yellow arrows indicate single tdTom+ cells at E12.5. **d** Long-term lineage tracing in the female reproductive tract induced from E11.5 to P90. Immunostaining for tdTomato reveals a major contribution of embryonic Lgr5+ cells to the epithelia of the adult oviduct, uterus and vagina. Red arrows indicate glandular cells. Scale bars, 50 μm. All images are representative of three independent mice.

(Supplementary Fig. 4a–e), and traced Lgr5+ cell-derived progeny to various ages. When tracing was initiated at P7, the *tdTomato* reporter gene was activated in both LE and developing GE, consistent with the location of endogenous Lgr5+ cells at this stage (Fig. 4a; 1 day post injection). One week later, multiple contiguous tdTom+ tracing units had expanded deep into the outer stromal area as endometrial glands developed (Fig. 4b; 7 days post injection). At P56, extensive dTom+ tracing units were readily evident throughout both the LE and GE endometrial compartments (Fig. 4c). Approximately 10% of the P7-P56 dTom+ cells co-expressed *Foxa2*, a specific marker of GE[20], indicating a long-term contribution of P7 Lgr5+ cells to the

development and maintenance of both the endometrial LE and GE (Fig. 4f, h). In contrast, when tracing was initiated from P14, the tdTomato reporter was predominantly activated at the tips of developing GE (Fig. 4d; 2 days post injection), reflecting the endogenous distribution of Lgr5high cells at this stage. At later time points, expanding tracing units were largely Foxa2+, consistent with a restricted contribution of P14 Lgr5high cells to GE development and maintenance (Fig. 4e, g, h and Supplementary Fig. 4f). These Lgr5 lineage-tracing data collectively indicate that endometrial Lgr5+ cells in prepubertal uterus are initially functioning as stem/progenitor cells responsible for the development of both the LE and GE, but gradually convert to being a

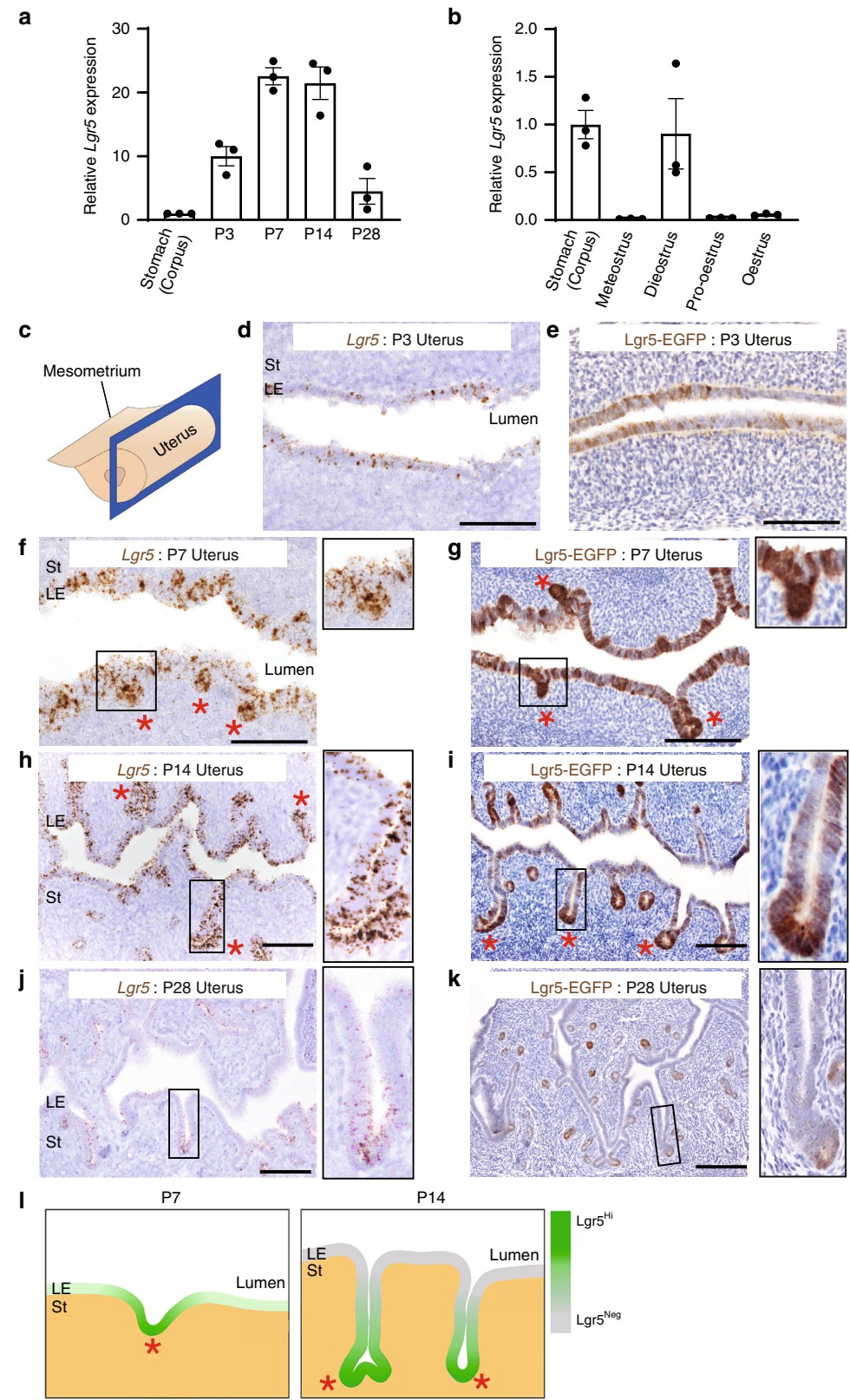

dedicated stem cell pool for the developing GE as the uterus matures. Together with the long-term tracing results, this suggests that the stem/progenitor cells in LE and GE exist independently around P14. In addition, co-IF for Lgr5-EGFP and Foxa2 in P14 uterus showed that endogenous *Lgr5* and

*Foxa2* expression is largely mutually exclusive, indicating that *Lgr5* expression in the endometrial glands is downregulated once the cells adopt a Foxa2+ glandular cell fate (Fig. 4i). Of note, Lgr5-driven lineage tracing was strictly confined to the K8+ epithelial compartment of the endometrium (Fig. 4j, k) and

**Fig. 3** Lgr5 is dynamically expressed in the developing uterus after birth. **a** QPCR analysis of *Lgr5* expression in uterus at various neonatal stages. Data from three independent mice are presented as mean ± s.e.m. **b** QPCR analysis for *Lgr5* on adult uterus (P90) from different estrous stages. Data from three independent mice are presented as mean ± s.e.m. **c** Cartoon depicting the tissue sectioning method. **d**, **f**, **h**, **j** RNA ISH for *Lgr5* on uterus from various ages during prepuberty period. **e**, **g**, **i**, **k** Immunostaining for EGFP in *Lgr5-2A-EGFP* mouse uterus at various ages. Red asterisks indicate Lgr5high cells at the tips of developing glandular epithelium (GE). Boxed enlargements highlight *Lgr5* expression within the developing glands at the different developmental stages. LE, luminal epithelium; St, stroma. Scale bars, 100 μm. All images are representative of three independent mice. **l** Cartoon depicting the endogenous *Lgr5* expression profiles at P7 and P14.

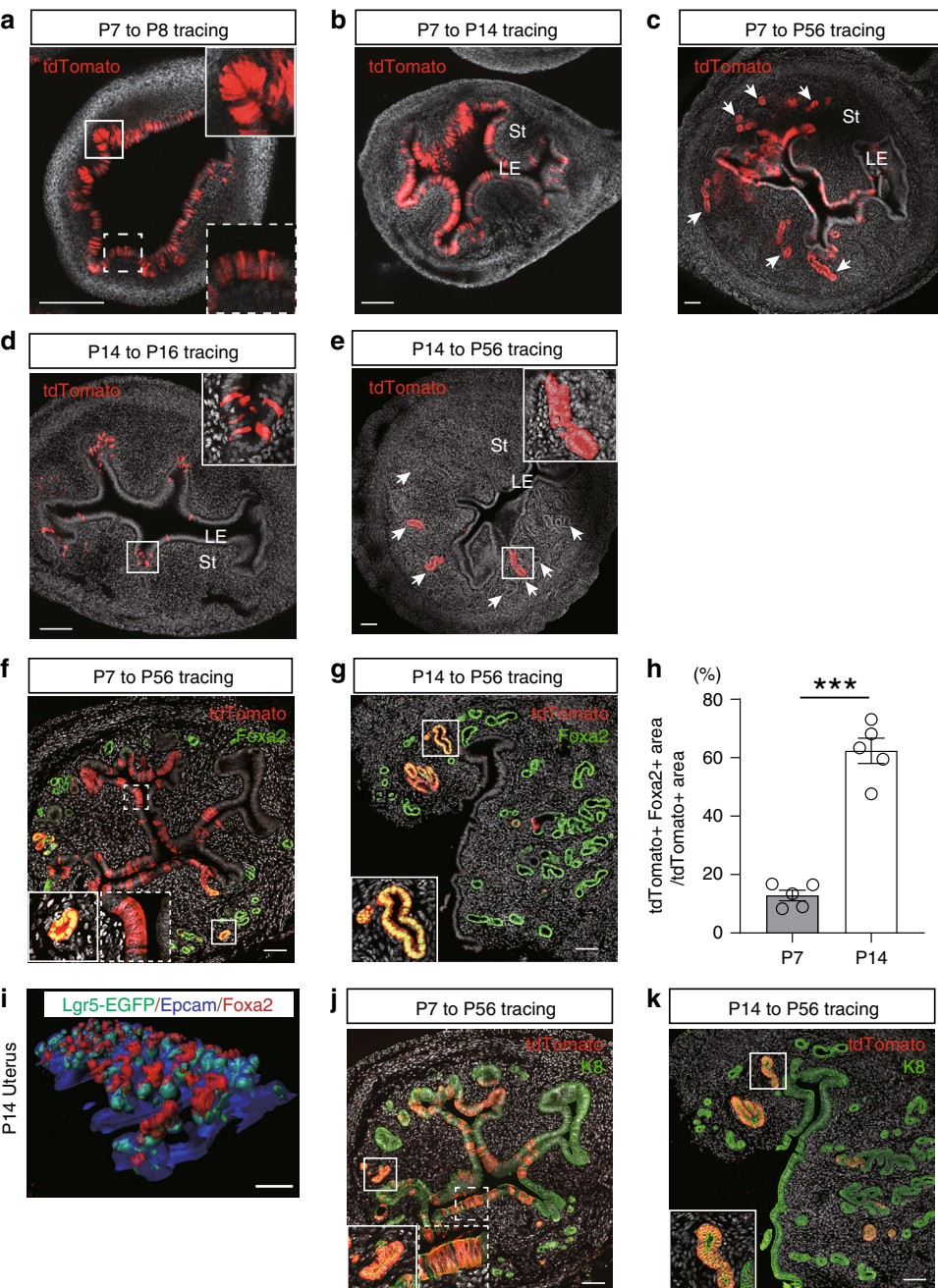

**Fig. 4** Pre-pubertal Lgr5⁺ populations are stem/progenitor cells for the developing uterus. **a–e** Endogenous tdTom fluorescence in *Lgr5-2A-CreERT2; R26-tdTomato* mouse uterus traced from P7 to P8 **a**, P7–P14 **b**, P7–P56 **c**, P14–P16 **d**, and P14–P56 **e**, respectively. Insets with solid and dashed lines indicate developing glandular epithelium (GE) and luminal epithelium (LE), respectively. White arrows indicate differentiated GE. St Stroma. **f**, **g** Co-IF for tdTom and Foxa2 in *Lgr5-2A-CreERT2; R26-tdTomato* mouse traced from P7 to P56 **f** and P14–P56 **g**. **h** Comparison of the percentage of Fox2⁺ cells within tdTom⁺ tracing units in P7–P56 and P14–P56 traced mice. Three independent fields from each mouse were analyzed. Data are represented as mean ± s.e.m. of the average of five independent mice. Data were tested for significance using unpaired two-tailed *t*-test (*P* = 9E−05, ***P* < 0.001). **i** 3D image of P14 *Lgr5-2A-EGFP* mouse uterus, co-stained for Epcam and Foxa2. **j**, **k** Co-IF for tdTom and K8 in *Lgr5-2A-CreERT2; R26-tdTomato* mouse traced from P7 to P56 **j** and P14–P56 **k**. Insets with solid and dashed lines indicate mature GE and LE, respectively. Scale bars, 100 μm.

was absent from the oviduct and upper vagina (Supplementary Fig. 2m–p).

To investigate any contribution of adult Lgr5+ cells to maintenance of the adult endometrium, we administered TAM to adult *Lgr5-2A-CreERT2; R26-tdTomato* mice at dioestrus stage. TdTom+ reporter gene activity was activated in scattered cells throughout the LE after 3 days (Supplementary Fig. 4g), as expected from the endogenous *Lgr5* expression pattern in dioestrus females (Supplementary Fig. 2b). To evaluate any contribution of adult Lgr5+ cells to estrus-driven endometrial epithelial renewal, we traced for a further 14 days, encompassing three complete estrus cycles. In contrast to the neonatal tracing results, there was a marked reduction in tdTom+ cells within the adult LE (Supplementary Fig. 4h). After 1 year of tracing, tdTom expression in the uterus was negligible, identifying adult Lgr5+ cells as likely being short-lived differentiated cells rather than self-renewing stem/progenitor cells contributing to adult LE homeostasis (Supplementary Fig. 4i).

**Lgr5+ cells are indispensable for uterine gland development**. To further explore the contribution of Lgr5+ cells to endometrial development, we employed the *Lgr5-DTR-EGFP* mouse model[21], which expresses an Lgr5+ cell-driven Diphtheria toxin (DT) receptor-EGFP fusion gene (Fig. 5a). The *DTR-EGFP* transgene reports endogenous *Lgr5* expression and facilitates selective ablation of *Lgr5*-expressing cells by administration of DT. *Lgr5-DTR-EGFP* mice were treated with DT at P7 and harvested at various ages (Fig. 5b). At 24 h post-injection, cleaved Caspase3-expressing cells were evident throughout the endometrium, including nascent glands, and *Lgr5-DTR-EGFP* expression was absent (Fig. 5c, Supplementary Fig. 5a, b), indicating efficient ablation of the endometrial Lgr5+ population. One week post-DT administration, there was a marked reduction in Foxa2+ glands compared with the wild-type control (Fig. 5d–f: P14), and the number of Foxa2+ glands was greatly reduced in mature uterus (Fig. 5h, i). In contrast, formation of Foxa2+ glands was not impaired by DT treatment of wild-type mice. Of note, the number of Foxa2+ glands is sustained in adulthood irrespective of estrous cycle, highlighting the validity of comparing gland numbers in mature uterus (Supplementary Fig. 5c, d). Treating *Lgr5-DTR-EGFP* mice with DT at P14 also resulted in a significant reduction of Foxa2+ glands (Supplementary Fig. 5e–g). Immunostaining for K8, progesterone receptor (PR), and Ki67 demonstrated normal LE formation and proliferation status, suggesting that ablation of Lgr5+ cells selectively abrogates GE development (Fig. 5g). Together, these results identify endometrial Lgr5+ cells as being essential for uterine gland development during the prepuberty period.

**Wnt activity of Lgr5high cells in the developing endometrium**. To further characterize endometrial Lgr5+ cells, we performed comparative gene expression profiling of fluorescence-activated cell sorting (FACS)-sorted GFP$^{high}$ versus GFP$^{neg}$ populations from endometrial cells (EPCAM+ cells) of P14 *Lgr5-2A-EGFP* mice (Fig. 6a, Supplementary Fig. 6a). Microarray analysis revealed 179 differentially expressed genes between the two populations (Fig. 6b, c). *Lgr5*, *Aldh1a1*, and *Prom1* were significantly higher in the GFP$^{high}$ population, whereas *Sprr1a* and *Zfp750* were enriched in the GFP$^{neg}$ cells. We further validated these results by qPCR and RNA ISH. *Lgr5* expression was highly enriched in the sorted GFP$^{high}$ population (five-fold), validating our sorting strategy (Fig. 6d). Another Wnt target gene, *Axin2*, was also enriched in the GFP$^{high}$ population (5.5-fold), indicating robust Wnt-signaling activity in this epithelial stem/progenitor population (Supplementary Fig. 6d, e). As expected from the

microarray analysis, *Aldh1a1* and *Prom1* were also markedly enriched in GFP$^{high}$ population (7.5-fold and 3-fold, respectively) in contrast to *Sprr1a* and *Zfp750*, which were strongly down-regulated in the Lgr5$^{high}$ compartment (Fig. 6d, Supplementary Fig. 6b). Independent RNA co-ISH analyses showed that while *Lgr5* expression colocalizes with that of *Aldh1a1 or Prom1*, it does not significantly overlap with *Sprr1* and *Zfp750* (Fig. 6e, Supplementary Fig. 6c). Notably, the expression of *Aldh1a1* has been reported to be upregulated in the glandular regions of neonatal endometrium[6,22], confirming that the Lgr5$^{high}$ endometrial population comprises predominantly GE cells.

Since various *Wnt* and *Fzd* genes are known to be important for uterus development, we next evaluated their relative expression levels in *Lgr5*-expressing cells. Analysis of microarray data showed distinct *Wnt/Fzd* expression patterns in GFP$^{high}$ and GFP$^{neg}$ populations (Fig. 6f). We confirmed the expression of essential Wnt/Fzd genes for uterus development by qPCR (Fig. 6g)[23-27]. Expression of *Fzd10*, a frizzled receptor expressed in developing uterine epithelium[23], was markedly higher in the GFP$^{high}$ population (3.1-fold). In contrast, *Wnt7b*, one of the Wnt ligands that potentially interact with Fzd10 to initiate Wnt signaling[27], was elevated in the GFP$^{neg}$ population (18-fold: Fig. 6g). Similarly, *Wnt4*, a well-established Wnt ligand essential for uterine development[24], was markedly higher in the GFP$^{neg}$ population. *Wnt7a*, another essential Wnt ligand that is expressed predominantly in the premature endometrium and is essential for uterine development[25], was expressed at similar levels between the two populations (Fig. 6g). Independent RNA co-ISH analyses showed that while *Lgr5* colocalizes with *Fzd10* expression, it does not significantly overlap with *Wnt7b* and *Wnt4* (Fig. 6h), confirming the qPCR data. Of note, *Wnt7a* is expressed broadly in the endometrium and does not correlate with *Lgr5* expression. *Wnt5a*, which is reportedly expressed in stroma[26], was selectively enriched in non-epithelial (EPCAM−) cells relative to both the GFP$^{high}$ and GFP$^{neg}$ epithelial populations (Supplementary Fig. 6f).

Lgr5 functions as a receptor for secreted R-spondins to modulate Wnt signal strength on Wnt-responsive stem cells in multiple tissues[28]. RNA ISH analysis of *R-spondin* expression revealed localized expression of *Rspo1* in stromal cells adjacent to the developing endometrium, including nascent glands harboring the Lgr5+ stem cells (Supplementary Fig. 6g). *Rspo3* expression was confined to the muscle layer, whilst Rspo2 and 4 were not detected.

These expression analyses indicate that epithelial Lgr5$^{high}$ cells in developing endometrium exhibit robust Wnt signaling that is potentially facilitated by receptors, such as *Fzd10* and *Wnt/R-spondin* ligands supplied by Lgr5$^{neg}$ cells in the adjacent endometrium or stroma.

**Lgr5high cells are Wnt-regulated endometrial stem cells**. To further define the identity and function of Lgr5+ cells in pre-pubertal endometrium, we exploited the near-physiological uterine organoid culture system[29,30]. Single EPCAM+GFP$^{high}$ cells sorted from P14 *Lgr5-2A-EGFP* mice efficiently generated organoids that exhibited a spherical phenotype (hereafter referred to as round-type organoids) in culture media supplemented with Rspo1 and Wnt3a (Fig. 7a, b). Importantly, the EPCAM+GFP$^{high}$ cell-derived organoids exhibited heterogenous expression of *EGFP* within the organoids, indicating a potential expansion of Lgr5+ stem cells (Fig. 7c). EPCAM+GFP$^{high}$ cells formed organoids at a significantly higher efficiency (8%) compared to EPCAM+GFP$^{neg}$ cells (<0.5%) (Fig. 7d, e). In addition, EPCAM+GFP$^{neg}$ cell-derived organoids did not survive beyond two passages, in contrast to EPCAM+GFP$^{high}$ cell-derived organoids that could be continually passaged in excess of 2 months (Fig. 7f).

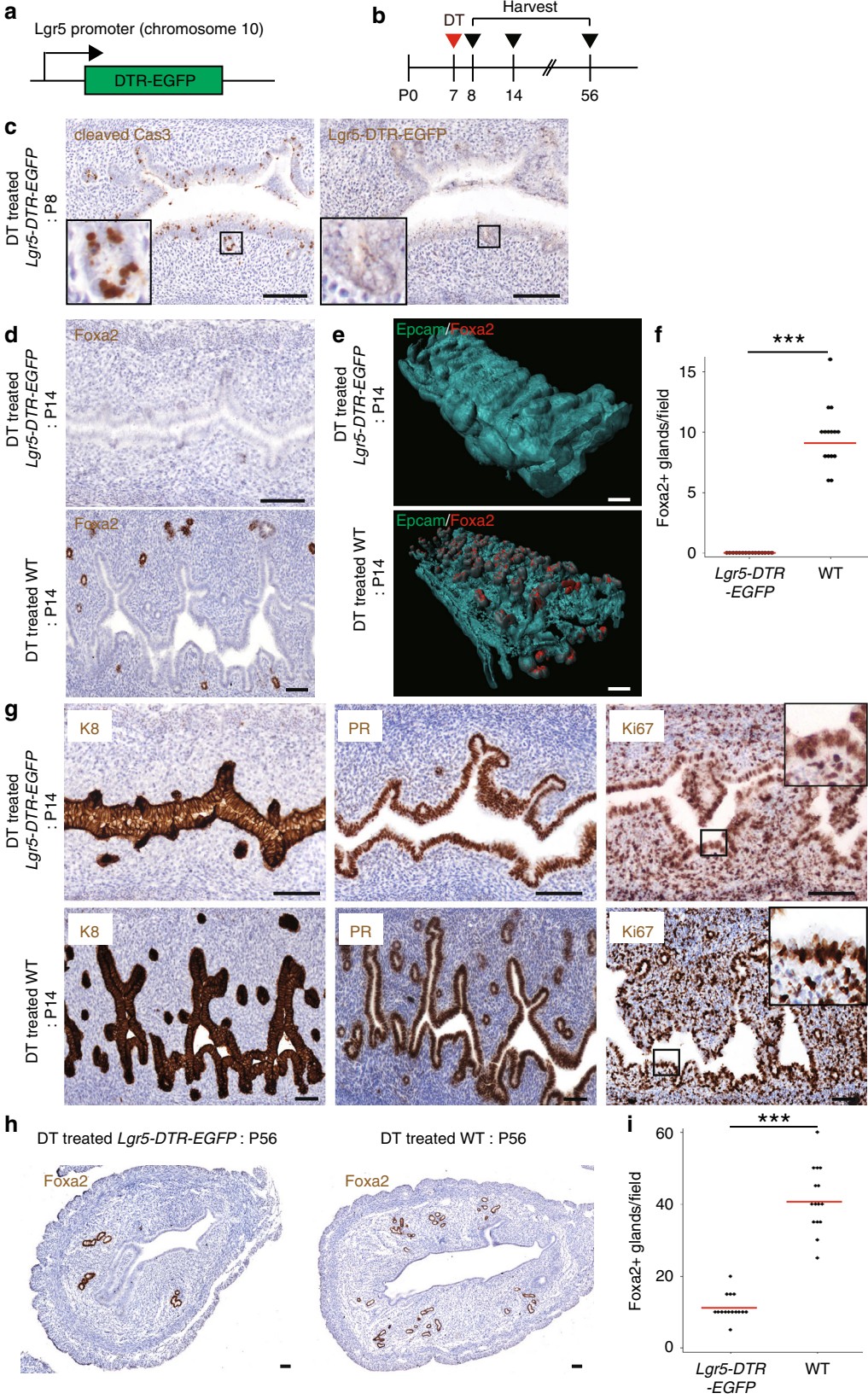

These results strongly indicate that only the EPCAM+GFP^high Lgr5+ endometrial cells function as a self-renewing stem/progenitor population in Rspo1/Wnt3a-supplemented ex vivo culture.

We next sought to ablate endogenous Lgr5+ cells in uterine organoids to evaluate their contribution to ex vivo organoid growth and maintenance. Uterine organoids were cultured for 3 days from EPCAM+ cells isolated from P14 *Lgr5-DTR-EGFP*

**Fig. 5** In vivo ablation of Lgr5+ cells in the developing uterus impairs gland formation. **a** The *Lgr5-DTR-EGFP* mouse model employed to ablate endogenous Lgr5+ cells. **b** Experimental strategy for ablating Lgr5+ cells in the developing uterus. **c** Immunostaining for cleaved caspase 3 and Lgr5-DTR-EGFP in DT-treated *Lgr5-DTR-EGFP* uterus at P8. **d** IHC for K8 and Foxa2 in DT-treated *Lgr5-DTR-EGFP* mouse and wild-type mouse (WT) at P14. **e** 3D images of P14 DT-treated *Lgr5-DTR-EGFP* and WT mouse, co-stained for Epcam and Foxa2. **f** Quantification of the number of Foxa2+ glands in DT-treated *Lgr5-DTR-EGFP* uterus and WT at P14. Three independent fields from each mouse were analyzed and the data from five independent mice are presented. Red bar represents median. Data were tested for significance using unpaired two-tailed *t*-test (*P* = 2E−09). **g** Immunostaining for K8, PR, and Ki67 in DT-treated *Lgr5-DTR-EGFP* and WT uterus at P14. **h** Immunostaining for Foxa2 in DT-treated *Lgr5-DTR-EGFP* and WT uterus at P56. **i** Quantification of the number of Foxa2+ glands in DT-treated *Lgr5-DTR-EGFP* and WT uterus at P56. Three independent fields from each mouse were analyzed and the data from five independent mice are presented. Red bar represents median. Data were tested for significance using unpaired two-tailed *t*-test (*P* = 2E−09). Scale bars, 100 μm. All images are representative of three independent mice per genotype. ***P < 0.001.

mouse uterus and DT was then added to ablate the resident Lgr5+ cells (Fig. 7g). We observed a ~50% reduction in outgrowth efficacy and an associated major decrease in organoid size in the DT-treated organoids compared to control (Fig. 7h, i). To evaluate whether the residual organoid growth observed using the *Lgr5-DTR-EGFP* model was due to incomplete Lgr5+ cell ablation, we repeated the experiment using our new, highly efficient *Lgr5-2A-DTR* mouse model (Supplementary Fig. 7a, b). Here, we observed a complete inhibition of organoid outgrowth following DT treatment, indicating that Lgr5+ cells are completely indispensable for organoid growth (Supplementary Fig. 7c, d). Of note, the high efficacy of this new *Lgr5-2A-DTR* ablation model precluded its use for in vivo ablation experiments in young mice due to rapid lethality resulting from systemic ablation of Lgr5+ stem cells essential for postnatal development.

Proliferating, progenitor-enriched organoids derived from Lgr5+ stem cells in other organs can be directed to differentiate by manipulating canonical Wnt-signaling levels[31]. To determine whether this is also true for immature uterus organoids, we established Lgr5+ cell-derived organoids in media supplemented with 80% less Wnt3a. We observed a marked phenotypic change in the organoids, with low Wnt conditions supporting the conversion of round-type organoids into vacuolated-type organoids (Fig. 7j, k). Pharmacological activation of canonical Wnt signaling by addition of CHIR99021, a GSK inhibitor, to the low Wnt3a condition media resulted in ~80% of the Lgr5+ cell-derived organoids adopting a round-type morphology (Fig. 7j, k). This suggested that the round-type organoids are more stem-like, while the vacuolated organoids are more differentiated. To better characterize the behavior of these two types of organoids, we first performed marker expression analysis. The round-type organoids exhibited a markedly higher proportion of Ki67-expressing cells than the vacuolated-type, indicative of their highly proliferative status (Fig. 7o). Expression of *Prss28*, a pre-differentiation marker, was expressed at similar levels in both organoid types (Fig. 7m). The vacuolated-type organoids displayed elevated expression of glandular differentiation markers *Foxa2* and *Lif*[22], reflecting their relatively differentiated status (Fig. 7n). As expected, qPCR analysis of organoids cultured with CHIR99021 revealed upregulation of the Wnt-signaling-associated components *Lgr5* and *Fzd10*, whilst the differentiation markers *Foxa2* and *Lif* were suppressed (Fig. 7l, m). In contrast, the LE markers *Wnt7a* and *Scnn1a* were expressed at similar levels in both organoid types while another LE marker *Irg1* was significantly higher in vacuolated-type organoids. However, expression of the various LE markers was markedly lower than in mature adult tissue, suggesting sub-physiological levels (Supplementary Fig. 7e).

Supplementing Wnt4 or Wnt7b in place of Wnt3a also activated *Axin2* and selectively promoted the generation of round-type organoids, as expected from their ability to upregulate *Lgr5* and *Fzd10* expression (Supplementary Fig. 7f–h). We then knocked down *Fzd10* in organoids to evaluate the functional relevance of *Fzd10* (Supplementary Fig. 7i, j). This resulted in a

major induction of apoptosis throughout the organoids, consistent with a role for *Fzd10* as an essential Wnt receptor in maintaining organoid growth ex vivo.

These observations highlight the role of Wnt signaling in regulating the balance between stemness/proliferation and glandular differentiation in immature endometrium derived from Wnt-dependent Lgr5high progenitor/stem cells.

**Lgr5neg endometrial epithelial cells are a Wnt niche**. Lgr5+ epithelial stem cells typically reside in close proximity to stromal/epithelial niche cells supplying secreted Wnt/R-spondin ligands and other key growth factors[32]. In the developing uterus, various essential Wnt ligands[5] and Rspo1 (Supplementary Fig. 6g) are expressed in the stromal compartment, and could act as a Wnt/R-spondin source for the Lgr5high stem cells. However, the robust *Wnt4/7b* expression we identified in the Lgr5neg epithelial cells also qualifies this endometrial population as a potential niche component for the Lgr5high stem cells during development. To explore this possibility, we first evaluated the impact of exogenous Wnt ligand reduction on organoid formation efficiency in vitro. EPCAM+GFPhigh cells sorted from P14 *Lgr5-2A-EGFP* mice could be developed into organoids under Wnt3a-reduced conditions ('Low Wnt3a') but the formation rate was markedly lower than in normal Wnt3a conditions (5% and 8%, respectively; Figs. 7e, 8c). We then co-cultured EPCAM+GFPhigh cells with EPCAM+GFPneg cells in the low Wnt3a condition (Fig. 8a). The organoid formation efficiency returned to similar levels observed using the normal Wnt3a condition (7.8%, Fig. 8b, c) and the mean size of generated organoids was greater under co-culture conditions ('+GFPneg') (Fig. 8d). Moreover, compared to the EPCAM+GFPhigh cell-derived organoids generated in low Wnt3a condition (Low Wnt3a), a significantly larger proportion (73% vs. 22%) of the organoids grown under co-culture conditions (+GFPneg) adopted the round-type morphology characteristic of a highly proliferative, undifferentiated state (Fig. 8b, e). When ETC159[33], a porcupine inhibitor, was initially added to the co-culture condition (+GFPneg) to inhibit any endogenous Wnt ligand secretion (+GFPneg +ETC159), the organoids predominantly appeared as vacuolated type (70%), and both the mean size and the organoid outgrowth efficacy were reduced (Fig. 8b–e). These results suggest that Lgr5neg epithelial cells expressing robust levels of *Wnt4/7b* can function as an endogenous Wnt source for Lgr5high stem cell-derived organoids in vitro.

To corroborate this finding in vivo, we evaluated neonatal uterine gland development following selective inhibition of paracrine Wnt signaling from epithelial cells by treatment of P7 wild-type mice with ETC159. After 7 days of daily i.p. injection, we evaluated glandular development by IHC (Fig. 8f). No Foxa2+ glands were observed in ETC159-treated mice, indicating that glandular development was significantly impaired by suppressing

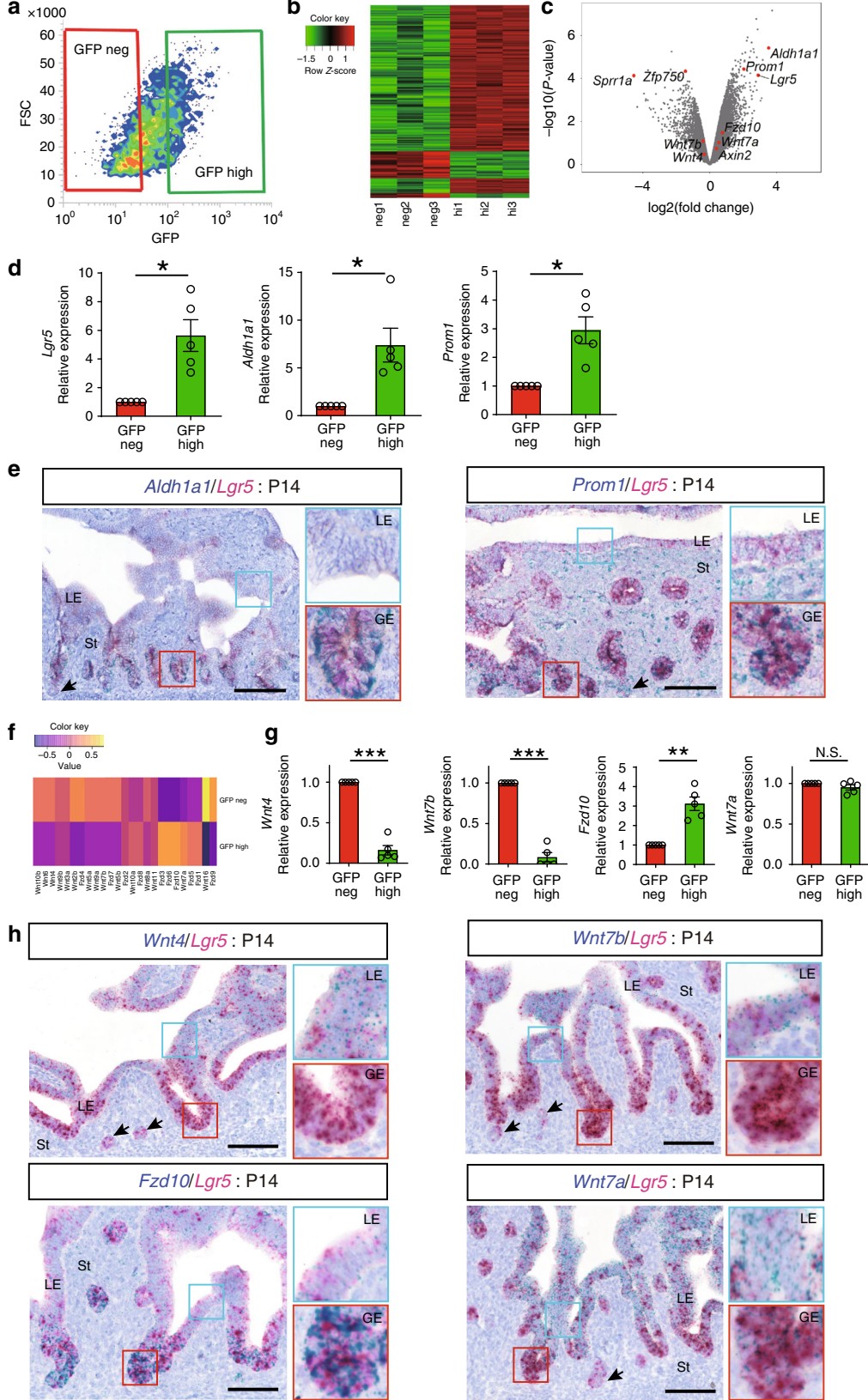

Wnt signaling in immature uterine epithelium (Fig. 8g–i). Notably, both *Lgr5* and *Axin2* expression in the ETC159-treated epithelium was markedly decreased and there was an accompanying major reduction in proliferation within the developing GE, confirming the efficient inhibition of Wnt

signaling in vivo (Supplementary Fig. 8a–c). In contrast, LE development was largely unaffected by ETC159 treatment.

Progesterone has been shown to be an inhibitor of gland development[34]. To evaluate whether *Lgr5* expression is impacted when gland development is perturbed, we treated neonatal mice

**Fig. 6** Gene expression profiling of Lgr5[high] cells in immature uterus. **a** Representative FACS profile of sorted cells from P14 *Lgr5-2A-EGFP* mouse uterus. **b** Heatmap of differentially expressed genes between GFP[high] and GFP[neg] cells. **c** Volcano plot from microarray analysis of all analyzed genes. **d** QPCR analysis for *Lgr5, Aldh1a1*, and *Prom1* on sorted EPCAM+GFP[high] cells. Data from five independent mice are presented as mean ± s.e.m. Data were tested for significance using unpaired two-tailed *t*-test (*Lgr5*: P = 0.013, *Aldh1a1*: P = 0.022, *Prom1*: P = 0.014). **e** RNA co-ISH of *Aldh1a1* and *Prom1* with *Lgr5* in P14 uterus. Insets with blue and red lines indicate LE and developing GE, respectively. Black arrows indicate differentiated GE. St Stroma. **f** Heatmap comparing *Wnt/Fzd* gene expression between GFP[high] and GFP[neg] cells. **g** QPCR analysis for *Wnt4, Wnt7b, Fzd10*, and *Wnt7a* on sorted EPCAM+GFP[high] cells. Data from five independent mice are presented as mean ± s.e.m. Data were tested for significance using unpaired two-tailed *t*-test (*Wnt4*: P = 1E−04, *Wnt7b*: P = 9E−05, *Fzd10*: P = 0.004, *Wnt7a*: P = 0.20). **h** RNA co-ISH of *Wnt4, Wnt7b, Fzd10*, and *Wnt7a* with *Lgr5* in P14 uterus. Insets with blue and red lines indicate LE and developing GE, respectively. Black arrows indicate differentiated GE. St, Stroma. Scale bars, 100 μm. All images are representative of five independent mice. ***P < 0.001, *P < 0.05, N.S., not significant.

with progesterone from P2 to P10. At P14, there was a significant reduction of *Lgr5* expression at developing gland buds (Supplementary Fig. 8d) indicating that the impairment in gland development by progesterone may be due to the reduction in *Lgr5* expression.

## Discussion

Here, we report the identification of mid-gestation embryonic Lgr5+ stem/progenitor cells contributing to the development of the female reproductive tract epithelia, and a population of neonatal Lgr5+ stem/progenitor cells supported by an epithelial Wnt source that are responsible for uterine gland development. Employing a new, non-variegated *Lgr5-2A-EGFP* reporter mouse model and highly sensitive RNA ISH techniques, we show that *Lgr5* expression in the developing female reproductive tract first originates in the E11.5 coelomic epithelium, the precursor of the Müllerian duct. In vivo lineage tracing with our non-variegated *Lgr5-2A-CreERT2; R26-tdTomato* mouse model identifies this coelomic Lgr5+ population as the origin of the epithelial lining of the upper vagina, uterus, and oviduct. We also observe robust contribution of embryonic Lgr5+ population to the ovary surface epithelium and granulosa lineages as previously reported[35]. Consistent with our previous report[35], *Lgr5* expression was detected in oviduct and upper vagina around birth, but rapidly declines to undetectable levels by P7. In contrast, *Lgr5* expression in the developing uterus exhibits a robust and dynamic expression pattern during pre-puberty. Prior to the onset of gland formation, *Lgr5* expression is uniformly distributed throughout the developing endometrium. However, coincident with the onset of adenogenesis around P7, *Lgr5* expression converts to a gradient, with highest levels present at the tips of developing antimesometrial glands. Fate mapping identifies these P7 Lgr5+ cells as stem/progenitor cells contributing to the development and maintenance of both the LE and GE compartments of the adult endometrium. *Lgr5* expression within the developing LE continues to decrease over the next 2–3 weeks, until it becomes confined to the tips of the maturing endometrial glands. Induction of Lgr5+ cell-driven lineage tracing at this stage identifies the gland-resident Lgr5[high] population as stem/progenitor cells contributing exclusively to the development and maintenance of the adult GE (Fig. 9). A previous study using a Pgr[Cre] conditional knockout mouse model to evaluate Lgr5 gene function in the neonatal endometrium demonstrated that Lgr5 was dispensable for normal uterine development[36]. Although Lgr5 function may be non-essential during these early stages, likely due to functional compensation by Lgr4, ablation of the *Lgr5*-expressing cells in our DT model demonstrates the significance and impact of the Lgr5+ progenitor cell population during uterus development. Somewhat surprisingly, LE development was largely unaffected by loss of the endogenous Lgr5+ cells, implying the existence of an Lgr5− stem/progenitor cells in this endometrial compartment at this stage. Collectively, these observations indicate that during pre-pubertal

development of the uterine epithelium, separate stem/progenitor pools are established for the GE and LE compartments. Lgr5+ cells serve as dedicated GE stem/progenitor cells in the maturing endometrium, whilst the identity of the LE stem/progenitor cells remains to be established. In the adult endometrium, *Lgr5* expression is confined to the LE, where it is hormonally regulated during the estrus cycle and appears to mark short-lived, non-stem/progenitor populations. It was recently reported that bipotent stem cells in adult endometrium reside in the zone between GE and LE, but further investigation is needed to better characterize them[37]. Given that the Lgr5+ cells are found in the developing GE residing proximally to Foxa2+ GE cells, it would be interesting to investigate whether the Lgr5+ postnatal progenitors can be the source of bipotent stem cells in the adult.

Lgr5+ stem cell populations in many tissues are regulated by canonical Wnt signaling, which is also known to be critical for uterine development in mice[38]. Comparative expression profiling of Lgr5+ stem/progenitor cells and their progeny in the developing endometrium revealed robust Wnt-signaling activity in the GE stem/progenitor cell compartment, likely transmitted via Fzd10 through interaction with several endogenous Wnt ligands, including Wnt4 and Wnt7b, which were expressed in adjacent Lgr5− cells. Given that *Aldh1a1* has been reported as a stem/progenitor marker in developing endometrium based on single RNA sequence analysis[6] and also *Prom1* is an established stem cell marker in small intestine[39], the upregulation of these genes in Lgr5+ cells further supports their stem cell identity. Lgr5 is a facultative component of the Wnt receptor complex, functioning to amplify canonical Wnt signaling on stem cell populations through interaction with its family of ligands, the R-spondins. Using sensitive ISH analyses, we document *Rspo1* expression within the stroma surrounding the Lgr5[high] stem/progenitor zone at the base of developing glands, highlighting a potential source of Wnt-modulating Lgr5 ligands in the pre-pubertal uterus.

Lgr5[high] cells FACS-sorted from developing endometrium supported Rspondin/Wnt-dependent outgrowth into epithelial organoids ex vivo, further underscoring the identity of the Lgr5[high] cells as a Wnt-regulated stem/progenitor population. Using this assay, we further show that canonical Wnt-signaling levels are instrumental in regulating the balance between stemness/proliferation and differentiation in immature endometrial organoids. Pharmacological inhibition of endogenous paracrine Wnt secretion in neonate mice using porcupine inhibitor established a similar dependence of uterine gland development on endogenous Wnt signaling, in agreement with similar phenotypes reported for various Wnt KO mouse models[24–26,38]. Conditional *Porcn* knockout mice exhibit variable uterus phenotypes, with complete ablation compromising only adult endometrial gland maintenance, in contrast to partial knockout, which suppresses proper gland formation[40,41]. The major impairment of endometrial gland formation we observe using pharmacological inhibition of Porcupine function, which targets the epithelial

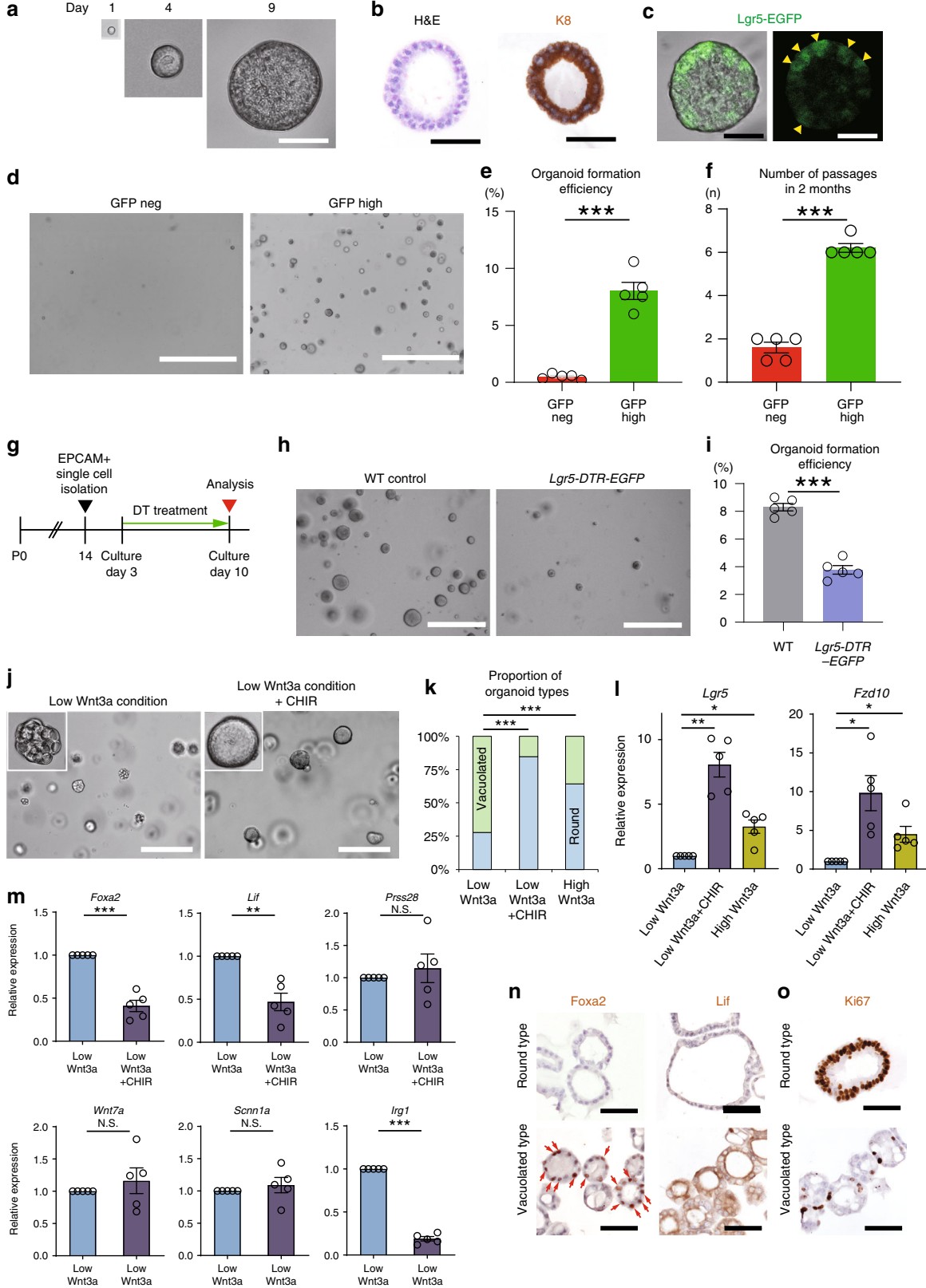

compartment[33], supports opposing roles for Wnt signaling within the stromal and epithelial compartments during neonatal gland formation. Importantly, we also identify Lgr5[−] endometrial cells as an epithelial Wnt source capable of substituting for exogenous Wnt ligands in supporting efficient organoid formation and expansion from Lgr5[high] cells ex vivo. Paneth cells have similarly

been identified as an epithelial Wnt source for Lgr5[+] stem cells in the intestine[42]. In future, it will be interesting to investigate the potential existence of functional redundancy between the epithelial and stromal Wnt sources in the developing uterus.

Progesterone is known to suppress endometrial development[34]. We show Lgr5 expression is down-regulated in progesterone-

**Fig. 7** Pre-pubertal Lgr5[high] uterus cells behave as Wnt-regulated stem/progenitor cells in ex vivo organoid culture assays. **a** Representative image of a developing uterine organoid generated from a single EPCAM+GFP[high] cell isolated from a P14 *Lgr5-2A-EGFP* uterus. **b** H&E and IHC for K8 on a Lgr5[high] cell-derived organoid. **c** Endogenous EGFP fluorescence on a Lgr5[high] cell-derived organoid. Yellow arrowheads highlight the heterogenous expression of Lgr5-EGFP within individual organoids. Scale bars, 50 μm. **d** Organoids generated from single EPCAM+GFP[high] cells and EPCAM+GFP[neg] cells (image representative of five replicates). Scale bars, 1000 μm. **e** Quantification of organoid formation efficacy from single EPCAM+GFP[high] and EPCAM+GFP[neg] cells. Data from five independent experiments are presented as mean ± s.e.m. Data were tested for significance using unpaired two-tailed *t*-test. *** *P* < 0.001. **f** Passage frequency of organoids derived from single EPCAM+GFP[high] and EPCAM+GFP[neg] cells. Data from five independent experiments are presented as mean ± s.e.m. Data were tested for significance using unpaired two-tailed *t*-test ($P = 7E-07$). **g** Experimental strategy to ablate Lgr5+ cells in vitro. **h** Organoid cultures derived from *Lgr5-DTR-EGFP* and wild-type (control) EPCAM+ uterus cells following Lgr5+ cell ablation by DT treatment in vitro (image representative of five replicates). Scale bars, 200 μm. **i** Outgrowth efficiency of organoids derived from *Lgr5-DTR-EGFP* and wild-type mouse EPCAM+ uterus cells following DT treatment in vitro. Data from five independent experiments are presented as mean ± s.e.m. Data were tested for significance using unpaired two-tailed *t*-test ($P = 4E-06$). **j** Organoids derived from wild-type EPCAM+ uterus cells cultured under low Wnt3a and low Wnt3a + CHIR conditions (image representative of five replicates). Scale bars, 200 μm. **k** The average proportions of organoid types under low Wnt3a, low Wnt3a + CHIR and high Wnt3a conditions. Data from five independent experiments are presented. Data were tested for significance by a chi-square test (Low Wnt3a vs. Low Wnt3a + CHIR: $P = 2E-17$, Low Wnt3a vs. High Wnt3a: $P = 2E-08$). **l** QPCR analysis of *Lgr5* and *Fzd10* expression on organoids generated under each culture condition. Data from five independent experiments are presented as mean ± s.e.m. Data were tested for significance using unpaired two-tailed *t*-test (*Lgr5*: Low Wnt3a vs. Low Wnt3a + CHIR: $P = 0.002$, Low Wnt3a vs. High Wnt3a: $P = 0.012$, *Fzd10*: Low Wnt3a vs. Low Wnt3a + CHIR: $P = 0.018$, Low Wnt3a vs. High Wnt3a: $P = 0.027$). **m** QPCR analysis of GE markers (*Foxa2, Lif, Prss28*) and LE markers (*Wnt7a, Scnn1a, Irg1*) expression on organoids generated under each culture condition Data from five independent experiments are presented as mean ± s.e.m. Data were tested for significance using unpaired two-tailed *t*-test (*Foxa2*: $P = 9E-04$, *Lif*: $P = 0.006$, *Prss28*: $P = 0.54$, *Wnt7a*: $P = 0.46$, *Scnn1a*: $P = 0.49$, *Irg1*: $P = 5E-06$). **n, o** Immunostaining for Foxa2, Lif, and Ki67 on round and vacuolated-type organoids. Scale bars, 50 μm. Red arrows indicate Foxa2-positive cells (image representative of three independent mice). ***$P < 0.001$, **$P < 0.01$, *$P < 0.05$, N.S., not significant.

treated mice, indicating that inhibition of glandular development by progesterone treatment also affects Lgr5 *expression*. This is compatible with the findings of Sun. et al., where progesterone stimulation resulted in Lgr5 down-regulation[13]. However, it is not clear whether loss of *Lgr5* expression is a cause or consequence of impaired gland development and it is not possible to directly correlate this with the attenuated gland formation caused by loss of Lgr5+ cells in our own study. Given that we now show that Wnt signaling is required for gland development, it is possible that downregulation of *Lgr5* expression by progesterone impairs Wnt-signaling initiation at the membrane of the gland progenitor cells, impacting their endogenous function.

The early establishment of cellular hierarchies is thought to be crucial for optimal neonatal development, particularly in tissues undergoing branching morphogenesis, such as the lungs[43] or pancreas[44]. The model we propose here for endometrial gland formation is similar to that of pancreas development, where a cellular hierarchy harboring Lgr5+ stem/progenitor cells at its apex is formed within nascent glands to ensure the development and maintenance of the adult glands.

The findings reported here deliver a major advance in our understanding of Wnt-driven gland formation in the developing uterus, identifying dedicated Wnt-regulated, gland-resident Lgr5+ stem/progenitor cells and supporting epithelial niche cells as critical orchestrators of endometrial development to ensure successful pregnancy in adults.

## Methods
**Mouse models**. *Lgr5-2A-EGFP* and *Lgr5-2A-DTR* mice were generated by homologous recombination in embryonic stem cells targeting the *2A-EGFP* and *2A-DTR* cassette, respectively, to the stop codon of *Lgr5*[13]. *Lgr5-2A-CreERT2* mice were described previously[13]. The *Rosa26 tdTomato* mice were purchased from Jackson Labs. *Lgr5-DTR-EGFP* mouse model has been previously described[22]. Vaginal smears were taken from adult females to determine the estrous cycle stages based on the presence and proportion of leukocytes, nucleated cells, and cornified cells[45]. Sex genotyping was performed when analyzing embryos. Genotyping primers are described in Supplementary Table 1. All animal experiments were approved by the Institutional Animal Care and Use Committee of Singapore. The experiments were not randomized, and there was no blinded allocation during experiments and outcome assessment.

**Animal experiments**. 4OHT (Sigma, H7904) and tamoxifen (Sigma, T5648) were dissolved in sunflower oil and injected i.p. at a single dose. 4OHT was injected to

P7 mice at 0.5 μg/g body weight and tamoxifen was injected to P14 and older mice at 0.15 mg/g body weight. For induction of lineage tracing from embryonic stages, 0.2 mg/g body weight of tamoxifen was injected to the pregnant female mice at E11.5. Timed pregnancies were staged relative to a vaginal plug that represents day 0.5 (E0.5). For DT injections, P7 *Lgr5-DTR-EGFP* or WT mice were i.p. injected with a single dose of 16.6 μg/kg body weight DT (Sigma, D0564) dissolved in PBS. For porcupine inhibitor experiments, ETC159 (gift from Prof. David M. Virshup, Duke NUS Medical School, Singapore) was dissolved in 50% PEG400 (v/v) in water and was injected s.c. daily at a dose of 10 mg/kg body weight. Progesterone (Sigma, PHR1142) was dissolved in sesame oil (Sigma, S3547) and injected s.c. daily at 50 μg/g body weight from P2 to P10.

**Cell dissociation and flow cytometry**. Uterine horns were harvested from mice and finely chopped using scalpel blades. Minced tissue was then incubated in chelation buffer (5.6 mM sodium phosphate, 8 mM potassium phosphate, 96.2 mM sodium chloride, 1.6 mM potassium chloride, 43.4 mM sucrose, 54.9 mM D-sorbitol, 1 mM dithiothreitol) with 5 mM EDTA, 2 mg/ml collagenase I (Worthington, LS004196) and 1 mM DTT at 37 °C for 1 h. Chelation buffer containing tissue was filtered through 100 μm filter mesh, and centrifuged at 720×g at 4 °C for 3 min. The pellet was resuspended in 1X TrypLE (Life Technologies, 12604) with DNaseI (0.8 U/μl) (Sigma, D4513) and incubated at 37 °C for 10 min with intermittent trituration for digestion into single cells. Digestion was quenched by dilution with cold HBSS buffer. The suspension was centrifuged at 720×g at 4 °C for 3 min. The pellet was resuspended in HBSS with 5% fetal bovine serum (FBS, Hyclone) and filtered through a 40 μm strainer. Cells were then stained with EPCAM antibody (APC-conjugated, Biolegend, 118214) at 1:200 dilution and sorted on a BD Influx Cell Sorter (BD Biosciences) after adding 1 μg/ml propidium iodine (Life Technologies). Cells were collected in RLT Plus buffer (Qiagen) for RNA extraction or HBSS with 2% FBS and 1% PenStrep (Gibco) for organoid culture.

**RNA isolation and qPCR**. RNA was isolated from tissue, organoids, or sorted cells using Trizol (Qiagen) or RNeasy Universal Plus Kit (Qiagen) according to the manufacturer's instructions. cDNA was generated using Superscript III (Life Technologies) according to the manufacturer's instructions. qPCR was performed in triplicate per gene for at least three biological replicates using GoTaq SYBR Green dye (Promega, A6002) according to the manufacturer's instructions. Relative quantification of gene expression was analyzed with Step One Software v2.1 (Applied Biosystems) using the ΔΔCT method with *Gapdh* as an endogenous reference. The qPCR primers are described in Supplementary Table 1.

**Microarray and analysis**. Hybridization and washing protocols were performed according to Origene instructions. RNA quality was first determined by assessing the integrity of the 28s and 18s ribosomal RNA bands on Agilent RNA 60000 Pico LabChips in an Agilent 2100 Bioanalyser (Agilent Technologies). A minimum of 2 ng of RNA was used to generate SPIA-amplified cDNA using the Ovation Pico WTA system (Nugen Technologies). Five micrograms of SPIA-amplified purified cDNA was then fragmented and biotin-labeled using the Nugen Encore Biotin module (Nugen Technologies). Microarray was performed using the Affymetrix Mouse ST v2.0 GeneChips (Affymetrix). The individual microarrays were washed

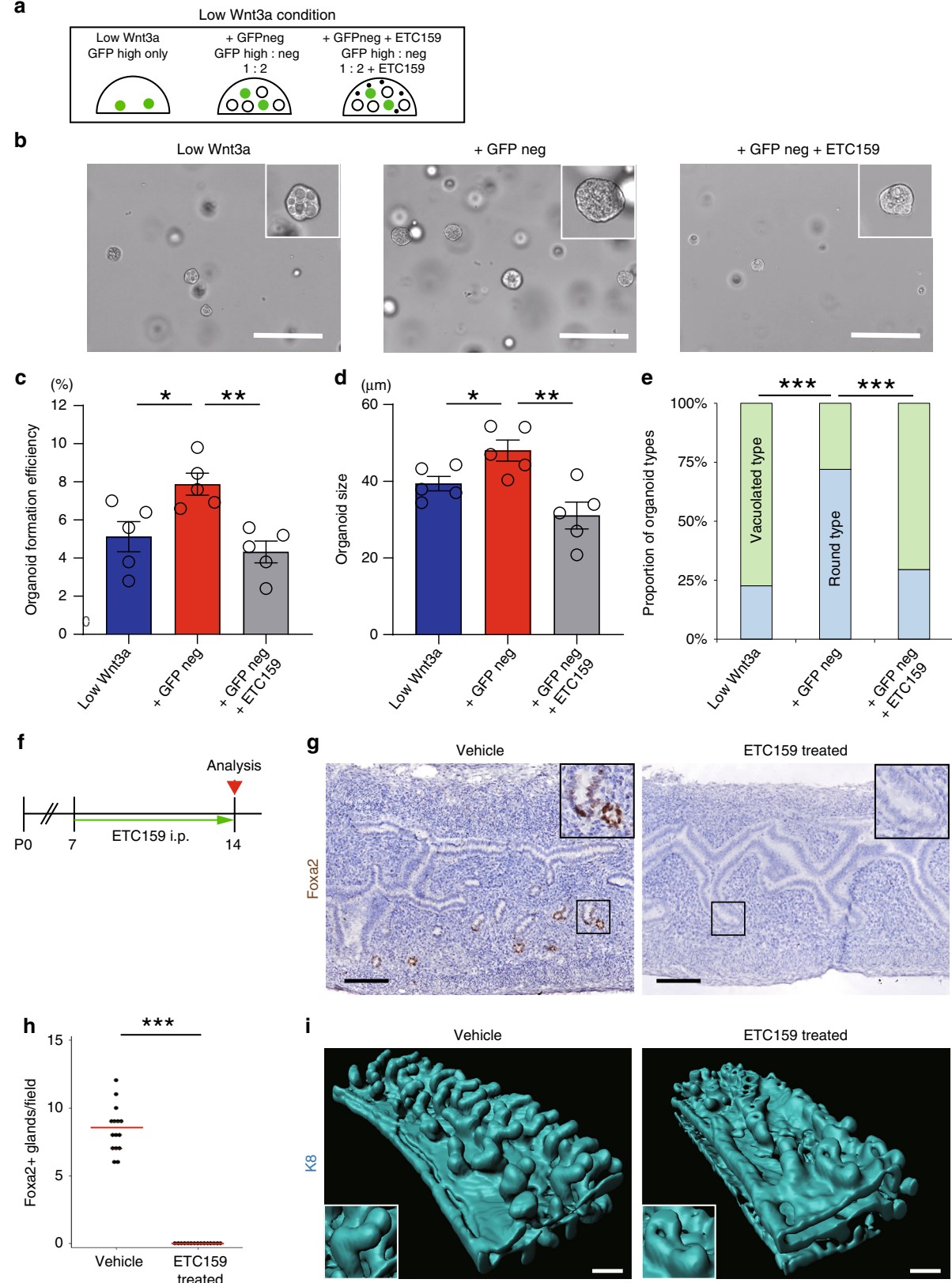

and stained in an Affymetrix Fluidics Station 450, and hybridized probe fluorescence was detected using the Affymetrix G3000 GeneArray Scanner. CEL files were generated for each array and used for gene expression analysis. The CEL files were then processed in R (v3.6.0) with the Bioconductor (v3.9) libraries oligo (v1.48.0), mogene20sttranscriptcluster.db (v8.7.0), and limma (v3.40.2). We used robust multi-array average (RMA)[46] to perform background correction and normalization

with the rma function implemented in the oligo package. The experimental design was stored as two factors with two levels each: batch (1 and 2) and Lgr5-EGFP level (high and low). Linear models were fitted to the expression data with the function lmFit, and differential expression was tested with eBayes (default parameters). P-values were adjusted using the Benjamini and Hochberg method. An adjusted P-value < 0.05 was considered as significant.

**Fig. 8** Defining an epithelial Wnt source for neonatal Lgr5[high] stem/progenitor cells in the developing uterus. **a** Experimental design of co-culturing EPCAM
+GFP[high] and EPCAM+GFP[neg] cells sorted from P14 *Lgr5-2A-EGFP* mouse. **b** Organoid cultures obtained using the indicated conditions (image
representative of five replicates). Scale bars, 200 μm. **c** The efficiency of organoid formation from single EPCAM+GFP[high] cells using the indicated culture
conditions. Data from five independent experiments are presented as mean ± s.e.m. Data were tested for significance using unpaired two-tailed *t*-test (Low
Wnt3a vs .+GFP neg: $P = 0.025$, +GFP neg vs. +GFP neg + ETC159: $P = 0.002$). **d** Quantitation of organoid size under each culture condition. Three
independent fields from each condition were analyzed. Data from five independent experiments are presented as mean ± s.e.m. Data were tested for
significance using unpaired two-tailed *t*-test (Low Wnt3a vs. +GFP neg: $P = 0.036$,+GFP neg vs. +GFP neg + ETC159: $P = 0.006$). **e** The average
proportions of organoid types generated using the indicated culture conditions from five independent experiments. Data were tested for significance by a
chi-square test (Low Wnt3a vs.+GFP neg: $P = 9E-13$, +GFP neg vs. +GFP neg + ETC159: $P = 8E-10$). **f** Experimental strategy of in vivo ETC159 treatment
to suppress Wnt secretion in neonatal uterus. **g** Immunostaining for Foxa2 on ETC159 treated ($n = 5$) and vehicle-treated control mouse ($n = 5$) at P14.
The insets with solid lines indicate developing GE. Scale bars, 200 μm. **h** Quantification of the number of Foxa2+ glands in ETC159- treated and vehicle-
treated control mouse uterus at P14. Three independent fields from each mouse were analyzed. Data from five independent mice are presented. Red bars
represent median. Data were tested for significance using unpaired two-tailed *t*-test ($P = 2E-11$). **i** 3D images of P14 vehicle and ETC159-treated WT
mouse, stained for K8. The insets show glandular areas. Scale bars, 100 μm. All images are representative of five independent mice per treatment. ***$P <$
0.001, **$P < 0.01$, *$P < 0.05$.

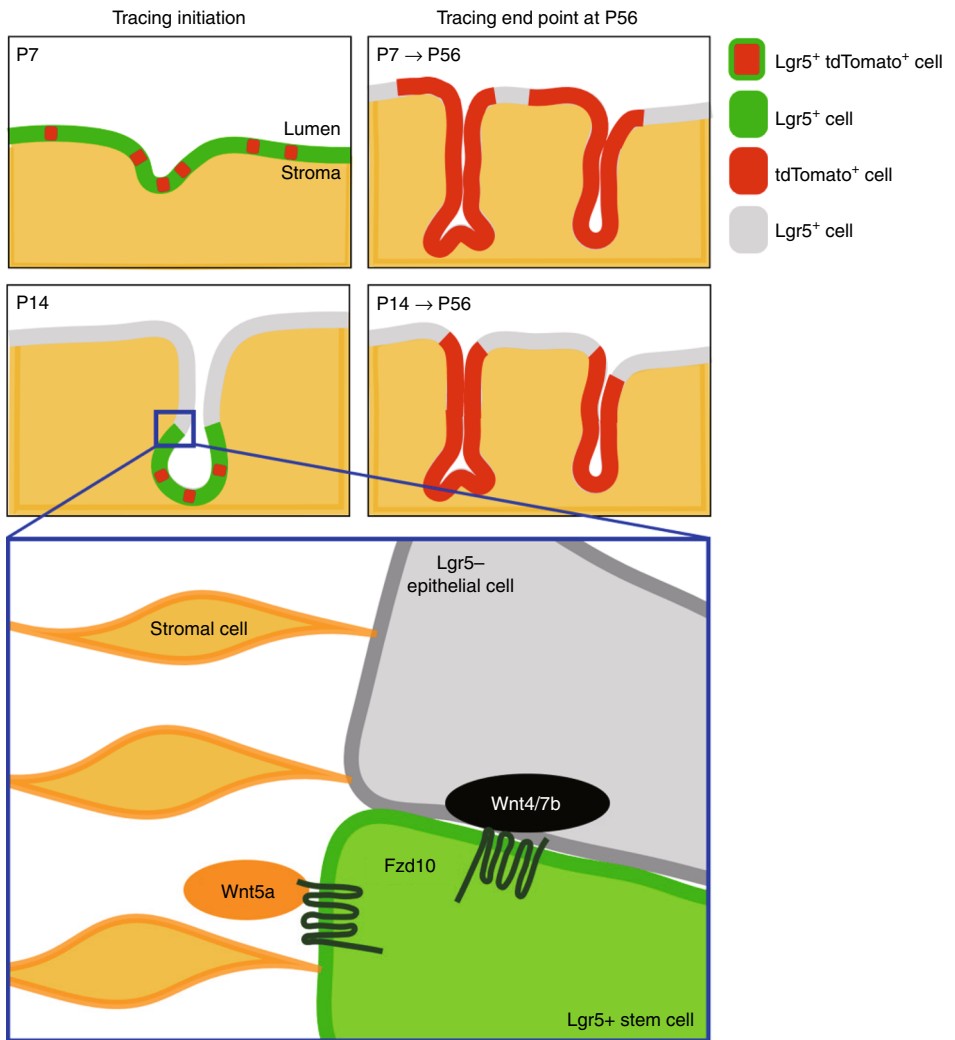

**Fig. 9** Summary cartoon depicting the dynamic expression pattern and function of Wnt-regulated Lgr5+ cells in the immature uterus. At P7, *Lgr5*+ cells are
uniformly distributed throughout the endometrium and contribute to the development of both the LE and GE. After P14, Lgr5+ cells become restricted to
the tips of developing glands and exclusively contribute to the development of GE. Wnt signaling is instrumental in regulating the activity of the Lgr5+
stem/progenitor cells at P14, likely mediated by interaction of Fzd10 with Wnt ligands supplied by both epithelial and stromal populations.

**Organoid culture**. FACS-sorted single endometrial cells were seeded in growth
factor-reduced Matrigel (Corning, 356231) and cultured in Advanced DMEM/F-12
media supplemented with 10 mM HEPES, 2 mM Glutamax, 1x N2, 1x B27 (all
Invitrogen), *N*-acetyl-cysteine (Sigma, A9165), Primocin (Invivogen, ant-pm1),
50 ng/ml EGF (Invitrogen, PHG0311), 100 ng/ml FGF10 (Peprotech, 100-18B),
100 ng/ml Noggin (Peprotech, 250-38-50), 100 ng/ml Wnt3a (Millipore, GF-160),

1 μg/ml R-spondin (Peprotech,120-38-500), 50 ng/ml HGF (Peprotech, 100-39H),
2 μM A83-01 (Tocris, 2939) and 10 mM nicotinamide (Sigma, N0636). 10 μM
Rock-inhibitor Y27632 (Sigma, Y0503) was also added to single-cell cultures.
Culture medium was changed every other day. After 7–10 days of culture, orga-
noids were harvested for analysis. DT was added at a concentration of 0.25 ng/μl to
the culture medium to induce Lgr5+ cell ablation at the indicated time point.

20 µM CHIR 99021 (Tocris, 4423) and 100 µM ETC159 were, respectively, added for differentiation experiments. Wnt4 (R&D, 6076) and Wnt7b (Abnova, 7477-P) were added at 100 ng/ml to substitute for Wnt3a in indicated experiments.

**shRNA treatment on organoids.** Fzd10 shRNA plasmids (TRCN0000422549, TRCN0000071871) were purchased from Sigma-Aldrich and an empty shRNA plasmid (pLKO.1 puro, #8453) was purchased from Addgene. Lentivirus particles were prepared according to the manufacturer's instruction. For lentivirus infection, uterus organoids were dissociated by 1X TrypLE and mixed with each lentivirus particles and polybrene (8 µg/ml). The mixture was centrifuged at 600×g, 32 °C, for 1 h and incubated at 37 °C for 6 h. After the incubation, cells were collected and embedded into matrigel. Next day, puromycin (1 µg/ml) was added into culture media to select shRNA-transfected cells. After 3 days, organoids were collected and Fzd10 knock-down efficiency was confirmed by qPCR. Immunostaining of cleaved Caspase-3 was performed according to standard protocol. After the puromycin selection, organoids were collected from Matrigel and fixed onto glass slides with 4% PFA for 15 min. Fixed cells were rinsed with 1xPBS and permeabilized in 5% FBS/0.3% TritonX-100/ 1xPBS for 60 min. Cells were incubated with rabbit anti-cleaved Caspase-3 (Asp175) primary antibody (1:400, Cell Signaling) for 16 h at 4 °C in 1X TrypLE. After 1x PBS wash, anti-rabbit Alexa488 antibody (1:200, Abcam) was used to visualize cleaved Caspase-3. Organoids were mounted with VECTASHIELD mounting medium with DAPI (Vector Laboratories, H-1500). Leica TSC SP8 confocal microscope was used for image acquisition.

**Immunohistochemistry and immunofluorescence.** IHC/IF was performed according to the standard protocol. Tissues were fixed in 4% paraformaldehyde (PFA) overnight at 4 °C before paraffin embedding. IHC/IF was performed on deparaffinized and rehydrated 6 µm tissue sections. Antigen retrieval was carried out by heating slides at 121 °C for 20 min using 2100 Antigen Retriever (Aptum Biologics) either in a modified citrate buffer, pH 6.1 (S1699, DAKO) or Tris/EDTA buffer, pH 9.0 (S2367, DAKO). The following primary antibodies were employed: mouse anti-Lim1 (1:100; DSHB, 4F2), rabbit anti-Foxa2 (1:400; Cell Signaling, 8186), rabbit anti-K8 (1:400; Abcam, ab53280), rabbit anti-vimentin (1:1000; Abcam, ab92547), mouse anti-E-cadherin (1:200; BD Transduction Laboratories, 610181), rabbit anti-cleaved Caspase3 (1:200; Cell Signaling, 9661), rabbit anti-Ki67 (1:200; ThermoFisher, MA5-14520), rabbit anti-LIF (1:200; Origene, TA321468), mouse anti-Ki67 (1:200; BD Transduction Laboratories, 550609), chicken anti-GFP (1:100; Abcam, ab13970), rabbit anti-GFP (1:200; Cell Signaling, 2956S), rabbit anti-RFP (1:200; Rockland, 600-401-379), mouse anti-RFP (1:100; Abcam, ab125244). The peroxidase-conjugated secondary antibodies used were mouse/rabbit EnVision + (DAKO) for HRP IHC or anti-chicken/rabbit/mouse Alexa 488/568/647 IgG (1:500; Invitrogen) for IF. IHC slides were mounted using DPX (Sigma 1.07979.0500) and IF slides were mounted using Hydromount (National Diagnostics, HS-106) with Hoescht as nuclear counterstain. Immunostainings were repeated on at least three tissue sections per tissue block. Only representative immunostainings were included in the manuscript.

H&E staining was performed on deparaffinized and rehydrated 6 µm tissue sections which were stained with Haematoxylin 2 (Richard-Allan Scientific, 7231L) followed by Scott's blue reagent (0.2% NaHCO3 (w/v), 2% MgSO4 (w/v) in water), then Eosin (Sigma, HT110132). Stained sections were dehydrated and mounted using DPX (Sigma 1.07979.0500).

**Whole-mount imaging and immunostaining.** For whole-mount experiments, tissues were fixed in 4% PFA overnight at 4 °C, followed by permeabilization in 2% TritonX-100/PBS overnight at 4 °C. Tissue sections (500 µm) were generated by vibrating microtome and cleared in RapiClear 1.52 (Sunjin Lab, Hsinchu City, Taiwan) according to the manufacturer's instructions[13]. Hoechst dye was used as nuclear counterstain. For immunostaining of whole-mount tissues for 3D imaging, tissues were fixed and permeabilized as described above followed by transferring into blocking buffer (5% NGS and 1% BSA in PBS) at 4 °C for 2 h. After washing tissues with washing buffer (0.3% NaCl in 1% PBST) at 4 °C, the following primary antibodies were employed: rabbit anti-K8 (1:100; Abcam, ab53280) rabbit anti-Foxa2 (1:100; Cell Signaling, 8186) rat APC-conjugated anti-EPCAM (1:100; Biolegend, 118214). After incubating at 4 °C for 3 days, tissues were washed with washing buffer followed by employing second antibodies (anti-rabbit/mouse/rat Alexa 488/568/647 IgG, 1:200; Invitrogen). Antibodies were all diluted in diluent buffer (2% NGS and 1% BSA in PBS). Tissues were then cleared with RapiClear 1.52 (Sunjin Lab) according to the manufacturer's instructions[11]. Hoechst dye was used as nuclear counterstain.

**RNA ISH.** RNA ISH and co-ISH was performed using RNAscope 2.5 High Definition Brown Assay and Duplex Reagent Assay (Advanced Cell Diagnostics), respectively. Consecutive sections hybridized for *DapB* as negative control were included for each RNAscope experiment.

**Microscopy imaging.** The following microscopes were used for image acquisition: Nikon Ni-E microscope/DS-Ri2 camera for IHC, H&E and RNAscope slides and an Olympus FV3000RS upright confocal microscope for IF slides and whole-

mount images. Cultured organoids were imaged with Olympus DP-27 camera on Olympus IX53-inverted microscope.

**Image processing and analysis.** All IF and 2D whole mount images were processed using ImageJ (NIH). Quantification of Foxa2-positive glands was performed by counting them in cross-section images taken from the middle of uterine horns in line with previous publications[40,47]. The surface area of overlapping Foxa2 and tdTomato stainings was measured by ImageJ (NIH). Whole-mount 3D rendering images were processed using Imaris 9.2.0 software (BITPLANE).

**Statistical analysis.** Statistical analyses were performed using GraphPad Prism, STATA15, or Microsoft Excel function. Data were expressed as mean ± SE. Statistical differences were determined using unpaired two-tailed $t$-test, one-way ANOVA or chi-square test. $P$-values of statistical significance are represented as ***$P$ < 0.001, **$P$ < 0.01,*$P$ < 0.05.

**Reporting summary.** Further information on research design is available in the Nature Research Reporting Summary linked to this Article.

## Data availability

Microarray data that support the findings of this study have been deposited in the Gene Expression Omnibus (GEO) under accession code GSE137974.

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

## Acknowledgements

We thank IMB-AMP and SBIC-Nikon Imaging Centre for imaging assistance, A*STAR SIgN for FACS sorting. We thank Prof. D.M. Virshup, Duke NUS Medical School, Singapore for providing ETC159 and F. Sauvage, Development of Molecular Biology, Genentech, South San Francisco, CA 94080, USA for providing the *Lgr5-DTR-EGFP* mice. N.B. is supported by the Agency for Science, Technology and Research (A*STAR) and the National Research Foundation (NRF) NRFI2017-03.

## Author contributions

R.S. designed, performed experiments, analyzed data, and wrote the manuscript. K.B.A.M. performed histology experiments. Y.S. generated *Lgr5-2A-EGFP* and *Lgr5-2A-DTR* mouse lines and performed FACS experiments. C.L., L.T.T., A.H., and S.H.T. performed mouse experiments. Y.I. and S.N. performed microarray analysis. H.I. constructed 3D images. K.M. and Y.Y. performed organoid experiments. E.W. generated *Lgr5-2A-EGFP* and *Lgr5-2A-DTR* mouse lines. N.B. supervised the project, generated *Lgr5-2A-EGFP* and *Lgr5-2A-DTR* mouse lines and wrote the manuscript. All figures are created by the author and the co-authors.

## Competing interests

The authors declare no competing interests.
