## [Peer Review File · Nature Communications]

Reviewers' Comments:

Reviewer #1:

Remarks to the Author:

NCOMMS-18-38451

Neonatal Wnt-dependent LGR5+ Stem/Progenitor Cells, Supported by Epithelial Niche Cells, are Essential for Uterine Gland Development

SUMMARY

This manuscript presents a series of experiments studying Lgr5-expressing cells in the developing female reproductive tract and uterus of the mouse. The studies utilize in vivo lineage tracing and ablation in mice as well as ex vivo organoids. The authors contend that the findings support the hypothesis that Lgr5-expressing cells in the Mullerian duct and uterine epithelium are stem cells for the uterine glands that develop in the neonate and that Lgr5 negative cells support the function of the Lgr5 positive stem cells.

OVERALL AND MAJOR COMMENTS

This manuscript addresses an interesting area of developmental biology. The authors present a series of studies using several different mouse models for in vivo lineage tracing and ablation experiments. Although the experiments are interesting, they are a bit confusing in terms of the effectiveness of the mouse models and interpretation of the results. Careful review generated a number of major comments.

- (1) The authors do not cite a 2014 paper (Sun et al., FASEB J 2014;28(5):2380-9) in which Lgr5 was conditionally deleted after birth in the uterus using the progesterone receptor (Pgr)-Cre model. In that model, Lgr5 was deleted in the uterine epithelium within the first 7 days after birth. However, no architectural differences were noted in the uterus, and they displayed normal epithelial development in terms of glands in the adult. Moreover, the mice were fertile when progesterone was supplemented due to a defect in ovarian function. The results of this paper conflict with the reviewed paper and are not addressed in the reviewed paper.
- (2) The mouse models used for lineage tracing and expression studies utilize fluorescent reporters (EGFP or tdTomato). For some studies, the authors present fluorescence data, but other studies present data for detection of the reporters using immunohistochemistry. It is not apparent why the authors switch between detection methods.
- (3) The histology of the Mullerian duct, uterus and other components of the female reproductive tract are presented at high magnification. Lower magnification photomicrographs should be presented for the reader to ascertain the type of section (transverse or longitudinal) as well as orientation (antimesometrial vs mesometrial). This presentation is particularly important to discriminate the luminal epithelium and glandular epithelium in the developing and adult uterus.
- (4) The use of a tamoxifen inducible reporter is a good approach for the embryo, however tamoxifen is xenoestrogen for the developing neonatal uterus. Tamoxifen can alter development of the uterine epithelium and induce pathology such as adenomyosis and adenocarcinoma.
- (5) Figure 4: Why are not all Foxa2-positive glands also Lgr5 positive in the lineage tracing studies if Lgr5-expressing cells are stem cells for the uterine glands?
- (6) The organoid experiments in Figures 7 and 8 are of particular interest and novel, but the generated organoids are not sufficiently well characterized to determine if they are predominantly luminal epithelium or glandular epithelium in terms of their phenotype. The Foxa2 IHC presented in Figure 7m is not convincing. The different types of organoids (round vs budded) need to be molecularly phenotyped to understand and interpret the results of the various studies.
- (7) The Materials and Methods are not sufficiently detailed to allow the reader to replicate the experiments.

SPECIFIC COMMENTS

Figure 2 Why is the Wolffian duct not labeled with tdTomato in Figures 2b and 2c as Lgr5 and Lim1 are expressed in that duct in Figure 1c? Why are Lgr5-expressing cells lineage traced to the luminal epithelium on P90 if they are only present in the uterine glands on P28 based on Figures 3i and 3j?

Figure 3 Panel a and b: What is the corpus? Panels c-j: Why is the EGFP detected more robustly than Lgr5 RNA?

Figure 5 The data with this mouse model DTR-EGFP are confusing given the recognized problems with the efficacy of the ablation noted by the authors.

Reviewer #2:

Remarks to the Author:

The uterine epithelium is one of the most regenerative tissues, undergoing cyclical morphologic and functional changes in response to fluctuating sex steroid hormones. Endometrial stem/progenitor cells are believed to contribute to the highly regenerative nature of this organ and their dysfunctions are suspected to lead to endometrial diseases including infertility, endometrial cancer, and endometriosis. Many endometrial stem/progenitor cells have been discovered in the past (Hum Reprod Update. 2016 PMID 26552890). However, there are very few studies using cell fate tracing approaches and so far, there is no definitive in-vivo evidence for the existence of endometrial stem cells. Bone-marrow-derived cells, stromal cells, and glandular epithelial cells are believed to contribute to endometrial regeneration. In this study, Seishima et al provide evidence that Lgr5+ cells are uterine epithelial stem cells in young but not in adult mice. The experiments are well performed and nicely documented. However, the definitive experiments supporting main arguments are not performed and at many places, the data presented doesn't support author claims (detailed below).

1. The Mullerian duct epithelium is mesoepithelial cells and co-express both mesenchymal (vimentin) and epithelial (pan-cytokeratin) markers. The Mullerian duct epithelium acquires more epithelial (organ-specific) features with development under the influence of segmental expression of Hoxa genes (Dev Biol 2007 Jun 15;306(2):493-504 17467685; Nat Rev Genet. 2003 Dec;4(12):969-80 14631357). If Lgr5-EGFP positive cells in the Mullerian duct epithelium (as shown in Fig. 1) are also positive for vimentin and negative for a more differentiated epithelial marker (E-Cadherin) then this doesn't mean these cells are mesenchymal cells. The data presented in Fig. 1 (few markers) is not sufficient to claim that the origin of Lgr5 expression in the developing reproductive tract as mesenchymal cells (page 6). To prove this, the author should pulse-label Lgr5+ mesenchymal cells before Mullerian duct formation and then perform lineage tracing experiments.

2. Lgr5 is expressed by all the Mullerian duct epithelial cells (Fig. 1e). As expected, genetic labeling of the Lgr5+ Mullerian duct epithelial cells results in the labeling of epithelia in oviduct, uterus, and vagina, the organs derived from the Mullerian duct. This is similar to lineage tracing of Wnt4+/Pax8+/Pax2+/Lhx1+/Wnt7a+/Lim1+ Mullerian duct epithelial cells (Hum Mol Genet. 2016 Mar 15;25(6):1059-73 26721931; Dev Biol. 2014 May 15;389(2):124-36 24560999; Development 2004 Feb;131(3):539-49. 14695376 and many others). The data presented in Fig. 2 shows that Mullerian duct epithelium gives rise to the epithelia of Mullerian duct-derived organs. There is not sufficient evidence to conclude that Lgr5+ cells in the Mullerian duct epithelium are stem/progenitor cells. The appearance of positive clones in Fig. 2 is due to insufficient recombination induced by tamoxifen. The authors might want to consider single cell seq to look for heterogeneity in the Mullerian duct epithelium followed by cell lineage tracing studies to prove this point.

3. Lgr5 expression is downregulated by estrogen (Figure 3 and Endocrinology 2009

Nov;150(11):5065-73 19797400). Even at the smallest doses (200ug), Tamoxifen and other similar estrogen agonists (such as Diethylstilbestrol/DES) exposure in prepubertal mice is able to induce uterine pathologies (Carcinogenesis 1997 Oct;18(10):2009-14. 9364013; Carcinogenesis 1997 Dec;18(12):2293-8. 9450472; Am J Obstet Gynecol. 1985 Mar 1;151(5):675-8. 3976766 and many others) partially by suppressing Wnt signaling (Nature Genetics 1998 Nov;20(3):228-30. 9806537). The authors are injecting tamoxifen to induce recombination in Lgr5^{cre} mice. It is likely that many of their observations are because of hyper-estrogenic activity of tamoxifen and not valid for normal WT mice uteri. Therefore, the authors should confirm some of these results (Fig. 4 and Suppl Fig. 4) using Lgr5-rtTA /tTA or other Wnt driven-rtTA/tTA mice.

4. Uterine glands are elongated, coiled tubular structures (Nature Communications 2018 Feb 9;9(1):603. 29426931; Mol Reprod Dev 2018 May;85(5):397-405. 29543367). 2D analysis is insufficient to claim that Lgr5 expression is limited to gland tips (Fig. 3h and j). As shown in panel 4j that Lgr5⁻ round glands are present adjacent to Lgr5⁺ glands. It is unclear if these are from the same gland or different glands. The author should perform 3D imaging on Lgr5-EGFP mice uteri to prove that Lgr5 expression is restricted to gland tips.

5. Previous studies from many different labs using 2D and 3D imaging have established that all the uterine glands are marked by Foxa2 (Nature Communications 2018 Feb 9;9(1):603. 29426931; PNAS 2017 Feb 7;114(6):E1018-E1026. 28049832). This study, for the first time, reports that Lgr5 positive glands are Foxa2 negative (Page 9 and Supp Fig. 4C). The author should perform 3D imaging on Lgr5-EGFP mice uteri to prove that such glandular heterogeneity exists. The authors might be looking at the same glands at a different depth which appear as different glands in 2D.

6. Ovarian hormones regulate Wnt signaling in the adult uterus. Expectedly, Lgr5, a Wnt target gene, expression also differs at the different stages of mouse estrus cycle (Figure 3 and Endocrinology 2009 Nov;150(11):5065-73 19797400). To examine whether Lgr5⁺ cells are stem cells in the adult uterus, the authors should first ovariectomized the mice to remove the influence of hormones on Lgr5 expression, by extension Lgr5^{cre}, and then activate the cre with Tamoxifen for cell fate tracing experiments. Later, hormone injections or pellets can be used to study these cells under the influence of ovarian hormones. Quantification should be provided for the number of cells labeled at the start and at different chase points. How these cells behave during artificial menstruation in mice (similar to Biol Reprod 2003 Oct;69(4):1273-80. 12801986) and during injury and repair (similar to Biol Reprod 2018 Sep 21. 30247509).

7. Figure 5 data show that Lgr5⁺ cells ablation before gland development leads to uteri with fewer glands. Histological sections presented are from different depths for control and mutants uteri giving a false impression that mutants have no Foxa2 positive glands. The authors should present thick sections with whole mount staining for Foxa2 or standard 3D imaging (similar to Nature Communications 2018 Feb 9;9(1):603. 29426931) to prove this. Also, what happens to glands if you ablate Lgr5⁺ cells after 15day post-natal when glands are already developed in mice?

8. Progesterone injections in neonatal mice suppress endometrial gland development (Biol Reprod 2012 May 10;86(5):146, 1-9. 22238285). Does progesterone treatment affect Lgr5⁺ stem cells to inhibit glandular development?

9. The authors claim that luminal epithelium is normal after the ablation of endometrial glands using Lgr-DTR-EGFP mice (page 10; Fig. 5d). However, provide no functional evidence for this claim (K8 staining is not enough). Endometrial organoids formation capacity is only limited to endometrial glands suggesting that stem cell activity is limited to the glandular epithelium (Development 2017 28442471). Do these remnant luminal epithelial cells develop organoids? Persists after injury? Proliferate? Respond to hormones?

10. The author should provide in-vivo data from Lgr5-2A-DTR ablation model (Fig. 7) by injecting DT directly to the uterine lumen (similar to Science 2011 18;331(6019):912-6. 21330545) to

avoid side effects on other organs.

11. Do these GFP low+ cells develop organoids (Fig. 6a)? The authors should characterize endometrial organoids derived from Lgr5+ cells. Expression of secretory, ciliated, and glandular markers? Why these organoids show no glandular budding similar to intestinal organoids? Lgr5+ cells should be cultured in the presence of Wnt4 and Wnt7a/b, the Wnt ligands expressed by the uterus, and not Wnt3a, not present in the uterus. This might explain the lack of glandular budding in these organoids. The organoids cultured in the low Wnt conditions are the vacuolated type, not budded type. Similar to the intestine, glandular buddings should be attached to the luminal epithelium.

12. To prove the concept of stem cell niche, the authors should use appropriate Wnt ligands (Wnt4 and Wnt7a/b) in their culture conditions. Conditioned media from Lgr5- cells is sufficient to replace Lgr5- cells?

13. Fig. 8 g ETC159 treated uterine section is close to utero-tubal junctions where there are naturally fewer glands and the vehicle-treated section from the middle of the uterus which is full of glands. The authors should provide 3D imaging or thick section whole mount imaging for this purpose.

14. In vivo, the epithelial cell-specific loss of Porcupine in mice is dispensable for glandular development but required for fertility (Biol Repro 2017 29036275). The ablation of Porcupine from both stromal and epithelial cells is also dispensable for endometrial gland development but required for their maintenance (Dev Biol 2017 27965056). Data presented in Fig. 8g is opposite to previous observations. ETC159 IP injections might have affected pituitary and ovarian functions, where Wnt plays a major role, to influence uterine gland development? OR uterine sections analyzed from different depths or locations have affected the assessment as glandular buds and extensions are clearly presented in both groups (Fig. 8g). Foxa2 is a direct target of Wnt signaling (Oncogene 2013 18;32(29):3477-82 22945641). Therefore, it is not appropriate to use Foxa2 as a proxy marker for the absence or presence of glands in Wnt suppressive conditions (Fig. 8g). Ck8 staining and 3D imaging is a more appropriate approach.

15. How Lgr5+ cells transcriptional profile differs from Lgr5- cells?

16. Data presented on the vagina is very distracting, given not much has been done on this organ. This data could be presented as supplementary data.

17. Current norm is at least 5 mice per time point compared to 3.

Reviewer #3:

Remarks to the Author:

The manuscript by Seishima and coworkers identifies the Lgr5+ cell in the developing uterus and uterine glands as stem/progenitor cell for the mouse uterus. This is a very significant and important finding. The work is thorough. The group used lineage marking, organoids and cell ablation as a means of demonstrating these cells are uterine glandular stem cells. This group also shows that it is paracrine WNT signaling that regulates the proliferation of these stem cells. The only issue that would be nice for this group to resolve is the co localization of these cells with Estrogen and Progesterone receptor since both regulate gland development. Other than one small issue this is an excellent manuscript.

Reviewers' comments:

Reviewer #1 (Remarks to the Author):

NCOMMS-18-38451

Neonatal Wnt-dependent LGR5+ Stem/Progenitor Cells, Supported by Epithelial Niche Cells, are Essential for Uterine Gland Development

SUMMARY

This manuscript presents a series of experiments studying Lgr5-expressing cells in the developing female reproductive tract and uterus of the mouse. The studies utilize in vivo lineage tracing and ablation in mice as well as ex vivo organoids. The authors contend that the findings support the hypothesis that Lgr5-expressing cells in the Mullerian duct and uterine epithelium are stem cells for the uterine glands that develop in the neonate and that Lgr5 negative cells support the function of the Lgr5 positive stem cells.

OVERALL AND MAJOR COMMENTS

This manuscript addresses an interesting area of developmental biology. The authors present a series of studies using several different mouse models for in vivo lineage tracing and ablation experiments. Although the experiments are interesting, they are a bit confusing in terms of the effectiveness of the mouse models and interpretation of the results. Careful review generated a number of major comments.

(1) The authors do not cite a 2014 paper (Sun et al., FASEB J 2014;28(5):2380-9) in which Lgr5 was conditionally deleted after birth in the uterus using the progesterone receptor (Pgr)-Cre model. In that model, Lgr5 was deleted in the uterine epithelium within the first 7 days after birth. However, no architectural differences were noted in the uterus, and they displayed normal epithelial development in terms of glands in the adult. Moreover, the mice were fertile when progesterone was supplemented due to a defect in ovarian function. The results of this paper conflict with the reviewed paper and are not addressed in the reviewed paper.

- We agree that the findings by Sun et al. should be addressed and cited in the manuscript. However, we respectfully disagree that our findings conflict with this study,

as our study is aimed at identifying a uterine epithelium progenitor cell population that is marked by Lgr5. This differs conceptually from Sun's conditional knock out model, which addresses the function of the Lgr5 Wnt co-receptor in the early stages of uterus development. The *in vivo* role of Lgr5 in modulating Wnt signal strength through its interaction with R-Spondin ligands is known to be redundant with similar functions of Lgr4 and Lgr6 (de Lau, W. Nature 2011). Given that Lgr4 is also expressed in the developing uterine epithelium (Supplementary Fig. 3d), it is not surprising that loss of Lgr5 expression alone does not impact on uterus development. In contrast, our cell ablation model directly evaluates the functional impact of Lgr5+ progenitor cell loss during uterus development rather than addressing Lgr5 receptor function in the developing uterus. To clarify this point, we have added the following.

(Page19, Line384-)

A previous study using a Pgr^{Cre} conditional knockout mouse model to evaluate Lgr5 gene function in the neonatal endometrium demonstrated that Lgr5 was dispensible for normal uterine development³⁶. Although Lgr5 function may be non-essential during these early stages, likely due to functional compensation by Lgr4, ablation of the Lgr5-expressing cells in our DT model demonstrates the significance and impact of the Lgr5+ progenitor cell population during uterus development.

(2) The mouse models used for lineage tracing and expression studies utilize fluorescent reporters (EGFP or tdTomato). For some studies, the authors present fluorescence data, but other studies present data for detection of the reporters using immunohistochemistry. It is not apparent why the authors switch between detection methods.

- We employed both whole-mount and IHC methodologies to optimally address different questions. We performed whole mount imaging to more broadly visualize endogenous EGFP or tdTomato expression in the developing uterus. We also utilized immunohistochemistry or immunofluorescence for marker quantitation or antibody co-staining. To aid readers in better visualizing the data, we now also present a 3D-reconstruction image from our whole-mount analyses (Fig. 4i).

(3) The histology of the Mullerian duct, uterus and other components of the female reproductive tract are presented at high magnification. Lower magnification photomicrographs should be presented for the reader to ascertain the type of section (transverse or longitudinal) as well as orientation (antimesometrial vs mesometrial). This presentation is particularly important to discriminate the luminal epithelium and glandular epithelium in the developing and adult uterus.

- Thank you for the constructive suggestion. Confocal z-stack image and cartoon illustrations have now been added to the figures to demonstrate tissue orientation (Fig. 1d, 3c). Accordingly, Fig. 3k has been replaced with a new image depicting an appropriate orientation.

(4) The use of a tamoxifen inducible reporter is a good approach for the embryo, however tamoxifen is xenoestrogen for the developing neonatal uterus. Tamoxifen can alter development of the uterine epithelium and induce pathology such as adenomyosis and adenocarcinoma.

- The reviewer is absolutely correct in stating that tamoxifen adversely affects the development and function of many estrogen-responsive tissues if applied at high dose, which can confound interpretation of *in vivo* lineage tracing results. We thank the reviewer for encouraging us to further investigate any impact of tamoxifen treatment on uterine development/function to further underscore the physiological significance of our findings. Here, to formally exclude significant adverse influence on uterus development/function, we carefully analyzed wild-type mice injected with varying doses of tamoxifen/4OHT at P7 or P14 by carefully quantifying the number of Foxa2+ glands at P56. As the ultimate readout of endogenous uterine functionality (and associated ovary function), we also bred the tamoxifen- or 4OHT-treated mice to directly evaluate their ability to carry to term. All data are shown in Supplementary Fig. 4a-c. We show that injection of 150mg/kg tamoxifen at P14, the dose used in the original manuscript, yielded comparable uterine gland numbers compared to untreated controls. When bred, these mice were able to consistently become pregnant and carry to term (n=5), indicating no significant effect of tamoxifen injection on either gland number or endogenous uterine function. However, injection of 100mg/kg tamoxifen at P7 did indeed negatively impact gland development based on a modest reduction in Foxa2+ gland numbers. Mice injected at this dose also had an associated inability to become pregnant, which may reflect a known impact of tamoxifen on hormonal regulation or implantation. Thus, we further tested different 4OHT doses and found that gland numbers were not significantly affected at 0.5mg/kg 4OHT, and the injected mice were able to consistently become pregnant and carry to term (n=4). To exclude the possibility of short-term effects of 4OHT on Lgr5 expression and proliferation, we examined Lgr5 and Ki67 expression in *Lgr5-2A-EGFP* mice at early time point (P8) following 0.5mg/kg 4OHT injection at P7, and found no significant differences compared with untreated control (Figures for revision a). We have since repeated all *in vivo* lineage tracing experiments from P7 using this new 4OHT dose, confirming the original results. This

new data is now included in Figs (4a, b, c, f, h, j). We are therefore fully confident that all original conclusions based on the Lgr5+ cell-derived lineage tracing data in the developing uterus are physiologically accurate.

(5) Figure 4: Why are not all Foxa2-positive glands also Lgr5 positive in the lineage tracing studies if Lgr5-expressing cells are stem cells for the uterine glands?

- Whole-mount analyses show that Foxa2+ population at the gland tip includes an Lgr5+/FoxA2+ subset adjacent to an Lgr5-/FoxA2+ subset, which is closer to the gland tip (Fig. 4i). For *in vivo* lineage tracing analyses, the requisite low doses of tamoxifen used in this study result in Cre-driven activation of the heritable reporter gene tdTomato in only a fraction of the Lgr5+ glands. This is evident in the short term tracing data shown in Figure 4a, d, where only limited numbers of Lgr5+ cells in a subset of the uterine gland tips express tdTomato soon after tamoxifen administration. Accordingly, only a fraction of the FoxA2-positive glands at later time-points demonstrate expansion of tdTomato expression due to endogenous Lgr5+ progenitor cell activity. Using higher doses of tamoxifen would increase the efficiency and frequency of the Lgr5-driven lineage tracing, resulting in many more traced glands, but as discussed above, this may negatively impact on uterine development/function, limiting the physiological accuracy of the data. The tracing frequencies here are fully in accordance with those observed in other Lgr5 stem cell-dependent tissues such as the intestine and stomach and, together with the organoid data, robustly support a stem/progenitor function for the endogenous Lgr5+ cells in the developing uterine glands.

(6) The organoid experiments in Figures 7 and 8 are of particular interest and novel, but the generated organoids are not sufficiently well characterized to determine if they are predominantly luminal epithelium or glandular epithelium in terms of their phenotype. The

Foxa2 IHC presented in Figure 7m is not convincing. The different types of organoids (round vs budded) need to be molecularly phenotyped to understand and interpret the results of the various studies.

- In response to reviewer's questions, we have further characterized the organoids in greater phenotypic and molecular detail when cultured under various Wnt conditions. Under normal Wnt conditions, organoids are spherical (referred to as round-type in text). When cultured in low Wnt conditions, organoids adopt an irregular shape consistent with the onset of cellular differentiation (as seen in other organoid systems, including intestine and stomach). Although they present a budding appearance, histological analysis reveals a phenotype that would more accurately be described as being vacuolated. We therefore altered this description accordingly throughout the text. More detailed characterization of these morphologically distinct organoid subtypes reveals markedly distinct phenotypes – the vacuolated-type organoids display limited proliferation and robust expression of glandular differentiation markers such as FoxA2 and Lif, in contrast to the round-type organoids, which are highly proliferative and express low levels of differentiation markers (Fig. 7m-o). The early differentiation marker Prss28 was expressed at low, but comparable levels in the two organoid sub-types as compared to adult tissue, likely reflecting partial commitment to some functional glandular lineages under both culture conditions. While expression of luminal markers Wnt7a, Scnn1a and Irg1 was detectable in both culture conditions (Fig. 7m), their levels were far below the physiologically relevant levels found in adult tissue (Supplementary Fig. 7e), likely underscoring the specificity of the culture conditions for sustaining glandular progenitor cell proliferation/differentiation *ex vivo*. Taken together, we conclude that round-type organoids comprise highly proliferative, undifferentiated progenitor cells, whilst vacuolated organoids occurring in low Wnt conditions resemble more highly differentiated glandular organoids. To clarify this point, we have added the following.

(Page14, Line294-)

This suggested that the round-type organoids are more stem-like while the vacuolated organoids are more differentiated. To better characterize the behavior of these two types of organoids, we first performed marker expression analysis. The round-type organoids exhibited a markedly higher proportion of *Ki67* expressing cells than the vacuolated-type, indicative of their highly proliferative status (Fig. 7o). Expression of *Prss28*, a pre-differentiation marker, was expressed at similar levels in both organoid types (Fig. 7m). The vacuolated-type organoids displayed elevated expression of glandular differentiation markers *Foxa2* and *Lif*²¹, reflecting their relatively differentiated status (Fig.

7n). As expected, qPCR analysis of organoids cultured with CHIR99021 revealed upregulation of the Wnt signaling associated components *Lgr5* and *Fzd10*, whilst the differentiation markers *Foxa2* and *Lif* were suppressed (Fig. 7l, m). In contrast, the LE markers *Wnt7a* and *Scnn1a* were expressed at similar levels in both organoid types while another LE marker *Irg1* was significantly higher in vacuolated-type organoids. However, the expression of the various LE markers were all markedly lower than in mature adult tissue, suggesting sub-physiological levels. (Supplementary Fig. 7e).

(7) The Materials and Methods are not sufficiently detailed to allow the reader to replicate the experiments.

- Thank you for pointing this out. We have now included additional experimental details and provided more information in the Materials and Methods.

SPECIFIC COMMENTS

Figure 2 Why is the Wolffian duct not labeled with tdTomato in Figures 2b and 2c as *Lgr5* and *Lim1* are expressed in that duct in Figure 1c?

- As reported elsewhere, the Wolffian duct begins to degenerate from E12.5 in female mice (Mullen RD et al. Sex Dev. 2014, Kobayashi A et al. Nat Rev Gen. 2003, Ma L. Trends Endocrinol Metab. 2009, Kurita T. Differentiation. 2011). We suspect that what we observed in our tracing model is related to the fact that degeneration of the Wolffian duct had already begun at the developmental time points in the manuscript. Although this is beyond the scope of our current study to explore in further detail, it would certainly be an interesting subject for future investigation.

Why are *Lgr5*-expressing cells lineage traced to the luminal epithelium on P90 if they are only present in the uterine glands on P28 based on Figures 3i and 3j?

- This long-term tracing data identifies the *Lgr5*⁺ cells present at P7/P14 as being neonatal stem/progenitor cells responsible for establishing and maintaining the adult glands. This approach traces the long-term output of the neonatal *Lgr5*⁺ cells – it is not dependent on, nor does it attempt to identify, *Lgr5*⁺ cells in the adult uterus. Given that *Lgr5*⁺ cells are also present at P28, initiating lineage tracing at this stage would likely also document a contribution to adult gland formation, but since we have already established this at earlier time-points, we consider it unnecessary to include these analyses.

Figure 3 Panel a and b: What is the corpus?

- The corpus refers to gastric corpus. We have changed the legend accordingly to represent this.

Panels c-j: Why is the EGFP detected more robustly than Lgr5 RNA?

- Endogenous *Lgr5* expression is typically maintained at low levels under homeostatic conditions, reflecting its important role in fine-tuning Wnt signaling on stem/progenitor populations. Visualization of this endogenous expression using in situ-hybridization is therefore technically challenging, especially given the short half-life of mRNA (5 mins). In contrast, the *Lgr5* reporter mice accumulate much higher levels of GFP protein in endogenous *Lgr5*⁺ cells due to its much longer half-life (1-2 hours; Barelle, C.J., et al. *Yeast* 2004, Chan L.Y., et al. *eLife* 2018), making it much easier to visualize. Nonetheless, there is an excellent correlation between *EGFP* expression and *Lgr5* mRNA in the developing uterus, highlighting the accuracy of the *Lgr5* reporter mice (Fig 3d-k), as previously shown for multiple other tissues including the intestine, stomach, skin, ovary and kidney.

Figure 5 The data with this mouse model DTR-EGFP are confusing given the recognized problems with the efficacy of the ablation noted by the authors.

- We apologize if we have not clearly conveyed our message here. In Fig 5, we use the well-validated *Lgr5*-DTR-EGFP ablation model to demonstrate a highly significant reduction in gland formation following a reduction in the number of endogenous *Lgr5*⁺ gland cells. Gland formation is not completely blocked in this model because the ablation efficiency of *Lgr5*⁺ cells is not 100% due to the fact that we can only use heterozygous animals (*Lgr5*-EGFP-DTR allele is a null allele and breeding to homozygosity is neonatal lethal), which consequently limits DTR expression levels on the *Lgr5*⁺ cells. We have recently generated a more efficient ablation model (*Lgr5*-2A-DTR), which leaves *Lgr5* function intact and can therefore be bred to homozygosity to maximize the sensitivity of *Lgr5*⁺ cells to DT treatment. For this reason, *Lgr5*⁺ cell ablation in this model is significantly more efficient than the *Lgr5*-DTR-EGFP model, which is what we demonstrate with *in vitro* organoid experiments in Supplementary Fig. 3c, d. Unfortunately, this new model is simply too efficient *in vivo*, causing rapid lethality due to the systemic ablation of all *Lgr5*⁺ stem cells in multiple tissues – this precludes its use for the *in vivo* ablation experiments in the uterus, which is why we employed the less efficient *Lgr5*-EGFP-DTR model for this purpose. The only difference between the models is the efficacy of *Lgr5*⁺ cell ablation, but the *in vivo*

phenotype obtained is very clear – even partial loss of the Lgr5+ gland cells has a major blockade on uterine gland development *in vivo*.

Reviewer #2 (Remarks to the Author):

The uterine epithelium is one of the most regenerative tissues, undergoing cyclical morphologic and functional changes in response to fluctuating sex steroid hormones. Endometrial stem/progenitor cells are believed to contribute to the highly regenerative nature of this organ and their dysfunctions are suspected to lead to endometrial diseases including infertility, endometrial cancer, and endometriosis. Many endometrial stem/progenitor cells have been discovered in the past (Hum Reprod Update. 2016 PMID 26552890). However, there are very few studies using cell fate tracing approaches and so far, there is no definitive *in-vivo* evidence for the existence of endometrial stem cells. Bone-marrow-derived cells, stromal cells, and glandular epithelial cells are believed to contribute to endometrial regeneration. In this study, Seishima et al provide evidence that Lgr5+ cells are uterine epithelial stem cells in young but not in adult mice. The experiments are well performed and nicely documented. However, the definitive experiments supporting main arguments are not performed and at many places, the data presented doesn't support author claims (detailed below).

1. The Mullerian duct epithelium is mesoepithelial cells and co-express both mesenchymal (vimentin) and epithelial (pan-cytokeratin) markers. The Mullerian duct epithelium acquires more epithelial (organ-specific) features with development under the influence of segmental expression of Hoxa genes (Dev Biol 2007 Jun 15;306(2):493-504 17467685; Nat Rev Genet. 2003 Dec;4(12):969-80 14631357). If Lgr5-EGFP positive cells in the Mullerian duct epithelium (as shown in Fig. 1) are also positive for vimentin and negative for a more differentiated epithelial marker (E-Cadherin) then this doesn't mean these cells are mesenchymal cells. The data presented in Fig. 1 (few markers) is not sufficient to claim that the origin of Lgr5 expression in the developing reproductive tract as mesenchymal cells (page 6). To prove this, the author should pulse-label Lgr5+ mesenchymal cells before Mullerian duct formation and then perform lineage tracing experiments.

- Thank you for this constructive comment. As we were only tracing a previously established fact and the epithelial/mesenchymal nature at this stage is not a main point of this study, we removed the vimentin stainings in embryonic stage tissues. Accordingly, we changed the description in the manuscript as following;

(Page 5, Line 86-)

At E12.5, during elongation of the Md, *Lgr5-EGFP* expression was maintained throughout the duct, as well as in Wd (Fig. 1d-f). At postnatal day 0 (P0), robust *Lgr5-EGFP* expression in the uterus was restricted to the epithelium, where it was broadly distributed (Fig. 1g). In contrast, *Lgr5-EGFP* was weakly expressed in oviduct or upper vagina, both of which originate from Md (Supplementary Fig. 1b, c). These data define the origin of *Lgr5* expression in the developing female reproductive tract as cells in the nascent Md. At the time of birth, expression is maintained within the epithelial lining of the developing uterus, as well as in oviduct and upper vagina.

2. *Lgr5* is expressed by all the Mullerian duct epithelial cells (Fig. 1e). As expected, genetic labeling of the *Lgr5*⁺ Mullerian duct epithelial cells results in the labeling of epithelia in oviduct, uterus, and vagina, the organs derived from the Mullerian duct. This is similar to lineage tracing of *Wnt4*⁺/*Pax8*⁺/*Pax2*⁺/*Lhx1*⁺/*Wnt7a*⁺/*Lim1*⁺ Mullerian duct epithelial cells (Hum Mol Genet. 2016 Mar 15;25(6):1059-73 26721931; Dev Biol. 2014 May 15;389(2):124-36 24560999; Development 2004 Feb;131(3):539-49. 14695376 and many others). The data presented in Fig. 2 shows that Mullerian duct epithelium gives rise to the epithelia of Mullerian duct-derived organs. There is not sufficient evidence to conclude that *Lgr5*⁺ cells in the Mullerian duct epithelium are stem/progenitor cells. The appearance of positive clones in Fig. 2 is due to insufficient recombination induced by tamoxifen. The authors might want to consider single cell seq to look for heterogeneity in the Mullerian duct epithelium followed by cell lineage tracing studies to prove this point.

- Whilst we agree that the *Lgr5*⁺ cell-derived lineage tracing data overlaps with similar data obtained using other genes expressed in the Mullerian duct, we respectfully disagree that this in any way invalidates our claim that *Lgr5*⁺ cells within the Mullerian duct are progenitor cells contributing to the development and long-term maintenance of the Mullerian-duct-derived organs. The appearance and maintenance of embryonic *Lgr5*⁺ cell-derived tracing units spanning the entire epithelium of the various adult reproductive organs is wholly consistent with a progenitor phenotype of the Mullerian duct *Lgr5*⁺ cells (this despite the fact that we only trace the output of a minor proportion of the Mullerian duct *Lgr5*⁺ cells due to the limited tamoxifen doses employed in this study to prevent tissue injury). In fact, we could further state that since lineage tracing is maintained long-term in these epithelia, the Mullerian duct *Lgr5*⁺ cells must have been the origin of the resident adult stem cell populations responsible for long term maintenance of these tissues. We do not attempt to claim that all of the embryonic *Lgr5*⁺ cells are functioning as stem/progenitor cells because we are acutely aware that the identity of stem/progenitor cells cannot be inferred on the basis of frequency, location or

phenotype and it is technically impossible to unequivocally demonstrate that each cell within a marker-positive population is a stem cell due to phenotypic heterogeneity arising from multilineage priming and cell intrinsic/niche-dictated proliferation/division modes. For the same reasons, we do not believe that analysis of any phenotypic heterogeneity within the Mullerian duct Lgr5+ population via single cell RNAseq would be informative in terms of further refining stem cell identity. Finally, as this constitutes a minor part of the paper, it would not justify the expensive, long-term investment required of the experiments suggested by the reviewer.

3. Lgr5 expression is downregulated by estrogen (Figure 3 and Endocrinology 2009 Nov;150(11):5065-73 19797400). Even at the smallest doses (200ug), Tamoxifen and other similar estrogen agonists (such as Diethylstilbestrol/DES) exposure in prepubertal mice is able to induce uterine pathologies (Carcinogenesis 1997 Oct;18(10):2009-14. 9364013; Carcinogenesis 1997 Dec;18(12):2293-8. 9450472; Am J Obstet Gynecol. 1985 Mar 1;151(5):675-8. 3976766 and many others) partially by suppressing Wnt signaling (Nature Genetics 1998 Nov;20(3):228-30. 9806537). The authors are injecting tamoxifen to induce recombination in Lgr5cre mice. It is likely that many of their observations are because of hyper-estrogenic activity of tamoxifen and not valid for normal WT mice uteri. Therefore, the authors should confirm some of these results (Fig. 4 and Suppl Fig. 4) using Lgr5-rtTA/tTA or other Wnt driven-rtTA/tTA mice.

- The reviewer is absolutely correct in stating that tamoxifen adversely affects the development and function of many estrogen-responsive tissues if applied at high dose, which can confound interpretation of *in vivo* lineage tracing results. We thank the reviewer for encouraging us to further investigate any impact of tamoxifen treatment on uterine development/function to further underscore the physiological significance of our findings. Here, to formally exclude significant adverse influence on uterus development/function, we carefully analyzed wild-type mice injected with varying doses of tamoxifen/4OHT doses at P7 or P14 by carefully quantifying the number of Foxa2+ glands at P56. As the ultimate readout of endogenous uterine functionality (and associated ovary function), we also bred the tamoxifen- or 4OHT-treated mice to directly evaluate their ability to carry to term. All data are shown in Supplementary Fig. 4a-c. We show that injection of 150mg/kg tamoxifen at P14, the dose used in the original manuscript, yielded comparable uterine gland numbers compared to untreated controls. When bred, these mice were able to consistently become pregnant and carry to term (n=5), indicating no significant effect of tamoxifen injection on either gland number or endogenous uterine function. However, injection of 100mg/kg tamoxifen at P7 did

indeed negatively impact gland development based on a modest reduction in FoxA2+ gland numbers, and an associated inability to become pregnant, which may reflect a known impact of tamoxifen on hormonal regulation or implantation. Thus, we further tested different 4OHT doses and found that gland numbers were not significantly affected at 0.5mg/kg 4OHT, and the injected mice were able to consistently become pregnant and carry to term (n=4). To exclude the possibility of short-term effects of 4OHT on Lgr5 expression and proliferation, we examined Lgr5 and Ki67 expression in Lgr52AeGFP mice at early time point (P8) following 0.5mg/kg 4OHT injection at P7, and found no significant differences compared with untreated control (Figures for revision a). We have since repeated all *in vivo* lineage tracing experiments from P7 using this new 4OHT dose, confirming the original results. This new data is now included in Figs (4a, b, c, f, h, j). We are therefore fully confident that all original conclusions based on the Lgr5+ cell-derived lineage tracing data in the developing uterus are physiologically accurate

4. Uterine glands are elongated, coiled tubular structures (Nature Communications 2018 Feb 9;9(1):603. 29426931; Mol Reprod Dev 2018 May;85(5):397-405. 29543367). 2D analysis is insufficient to claim that Lgr5 expression is limited to gland tips (Fig. 3h and j). As shown in panel 4j that Lgr5- round glands are present adjacent to Lgr5+ glands. It is unclear if these are from the same gland or different glands. The author should perform 3D imaging on Lgr5-EGFP mice uteri to prove that Lgr5 expression is restricted to gland tips.

- We thank the reviewer for this useful suggestion. We have now conducted 3D imaging of the neonatal uterus of an Lgr5-EGFP reporter mouse (Figure 4i). This nicely demonstrates that Lgr5^{high} cells are limited to the anti-mesometrial side (also from SFig. 3b), where Wnt signaling is reportedly highly activated (Goad, J., et al. Dev. Biol. 2017).

5. Previous studies from many different labs using 2D and 3D imaging have established that

all the uterine glands are marked by Foxa2 (Nature Communications 2018 Feb 9;9(1):603. 29426931; PNAS 2017 Feb 7;114(6):E1018-E1026. 28049832). This study, for the first time, reports that Lgr5 positive glands are Foxa2 negative (Page 9 and Supp Fig. 4C). The author should perform 3D imaging on Lgr5-EGFP mice uteri to prove that such glandular heterogeneity exists. The authors might be looking at the same glands at a different depth which appear as different glands in 2D.

- Thank you for this suggestion. We performed 3D imaging of neonatal uterus from an Lgr5-EGFP reporter mouse stained with FoxA2. Data are shown in Figure 4i. This analysis revealed the Foxa2⁺ population at the gland tip to comprise an Lgr5⁺/FoxA2⁺ subset adjacent to an Lgr5⁻/FoxA2⁺ subset, which is closer to the gland tip, indicating that *Lgr5* expression in the endometrial glands is downregulated once the cells adopt a Foxa2⁺ glandular cell fate. This description is included in the manuscript.

(Page9, Line171-)

In addition, co-IF for Lgr5-EGFP and Foxa2 in P14 uterus showed that endogenous *Lgr5* and *Foxa2* expression is largely mutually exclusive, indicating that *Lgr5* expression in the endometrial glands is downregulated once the cells adopt a Foxa2⁺ glandular cell fate (Fig. 4i).

6. Ovarian hormones regulate Wnt signaling in the adult uterus. Expectedly, Lgr5, a Wnt target gene, expression also differs at the different stages of mouse estrus cycle (Figure 3 and Endocrinology 2009 Nov;150(11):5065-73 19797400). To examine whether Lgr5⁺ cells are stem cells in the adult uterus, the authors should first ovariectomized the mice to remove the influence of hormones on Lgr5 expression, by extension Lgr5^{cre}, and then activate the cre with Tamoxifen for cell fate tracing experiments. Later, hormone injections or pellets can be used to study these cells under the influence of ovarian hormones. Quantification should be provided for the number of cells labeled at the start and at different chase points. How these cells behave during artificial menstruation in mice (similar to Biol Reprod 2003 Oct;69(4):1273-80. 12801986) and during injury and repair (similar to Biol Reprod 2018 Sep 21. 30247509).

- We performed short-term and long-term lineage tracing from adult mice to document that random Lgr5⁺ luminal cells do not expand or persist long term (1 year) (Supplementary Fig. 4g). These results suggest that Lgr5⁺ adult cells do not contribute to endometrial homeostasis, at least during normal estrous cycles. Exploring whether adult Lgr5⁺ cells can potentially function as endometrial stem cells in mouse models of human menstruation would be an interesting and exciting subject of future investigation, but is beyond the focus and scope of the current study, which focuses on uterine gland development.

7. Figure 5 data show that Lgr5+ cells ablation before gland development leads to uteri with fewer glands. Histological sections presented are from different depths for control and mutants uteri giving a false impression that mutants have no Foxa2 positive glands. The authors should present thick sections with whole mount staining for Foxa2 or standard 3D imaging (similar to Nature Communications 2018 Feb 9;9(1):603. 29426931) to prove this. Also, what happens to glands if you ablate Lgr5+ cells after 15day post-natal when glands are already developed in mice?

- We appreciate this helpful suggestion. We performed 3D imaging to document the global impact of Lgr5+ cell ablation on uterine gland formation using Lgr5-DTR-EGFP mice (Fig. 5e). This analysis further highlights the major reduction in Foxa2-positive gland numbers following Lgr5+ cell ablation *in vivo* at P7. Furthermore, in response to the reviewer query, we employed the same approach to demonstrate a comparable attenuation of normal gland formation following ablation of Lgr5+ cells at P14, 15 (Supplementary Fig. 5f, g).

8. Progesterone injections in neonatal mice suppress endometrial gland development (Biol Reprod 2012 May 10;86(5):146, 1-9. 22238285). Does progesterone treatment affect Lgr5+ stem cells to inhibit glandular development?

- As shown in Supplementary Figure 8d, *Lgr5* expression was downregulated in progesterone-treated mice, indicating that inhibition of glandular development by progesterone treatment also influences *Lgr5* expression. This is compatible with the findings of Sun, X. et al., showing that progesterone stimulation results in *Lgr5* down-regulation (Sun, X., et al. Endocrinology 2009). However, it is not clear whether loss of *Lgr5* expression is a cause or consequence of impaired gland development and it is not possible to directly correlate this with the attenuated gland formation caused by loss of Lgr5+ cells in our own study. However, given that we now show that Wnt signaling is required for gland development, it is possible that downregulation of *Lgr5* expression by progesterone impairs Wnt signaling initiation at the membrane of the gland progenitor cells, blocking their endogenous function. We now discuss this in the revised manuscript on page20.

(page20 Line433-)

Progesterone is known to suppress endometrial development³³. We show *Lgr5* expression is down-regulated in progesterone-treated mice, indicating that inhibition of glandular development by progesterone treatment also affects *Lgr5* expression. This is compatible with the findings of Sun, X. et al., where progesterone stimulation resulted in

Lgr5 down-regulation¹². However, it is not clear whether loss of *Lgr5* expression is a cause or consequence of impaired gland development and it is not possible to directly correlate this with the attenuated gland formation caused by loss of *Lgr5*⁺ cells in our own study. Given that we now show that Wnt signaling is required for gland development, it is possible that downregulation of *Lgr5* expression by progesterone impairs Wnt signaling initiation at the membrane of the gland progenitor cells, impacting their endogenous function.

9. The authors claim that luminal epithelium is normal after the ablation of endometrial glands using *Lgr*-DTR-EGFP mice (page 10; Fig. 5d). However, provide no functional evidence for this claim (K8 staining is not enough). Endometrial organoids formation capacity is only limited to endometrial glands suggesting that stem cell activity is limited to the glandular epithelium (Development 2017 28442471). Do these remnant luminal epithelial cells develop organoids? Persists after injury? Proliferate? Respond to hormones?

- To better characterize the luminal epithelium following *Lgr5*⁺ cell ablation, we additionally analyzed *PgR* expression status. We found no major difference in expression of this luminal marker between *Lgr5*⁺ cell-ablated (gland impaired) and wild-type uterus, confirming the lack of any significant effect on luminal cells (Fig. 5g). We also did not observe any change in its proliferation status following *Lgr5*⁺ cell ablation, indicating that the luminal epithelium does not respond to gland loss *in vivo*.
- We also addressed this issue using *in vitro* organoid cultures generated from DT-treated (gland-depleted) uterus. Organoid formation efficiency from *Lgr5*⁺ cell-ablated uterus epithelium was markedly decreased relative to wild-type (Figure for revision d, e, f). However, since we do not know whether the culture system supports the growth of luminal epithelium (and information on growth factor requirements of luminal differentiation are currently unknown), it is not possible to determine whether the observed reduction in organoid forming capacity in any way reflects an associated functional impact on the luminal epithelium. It is also true to say that our tracing/ablation data indicates that the glandular/luminal epithelial compartments are likely to be maintained by independent stem/progenitor populations, with very different growth (niche) requirements, precluding the generation of a single culture system that supports the growth of both epithelial compartments.

10. The author should provide in-vivo data from *Lgr5-2A-DTR* ablation model (Fig. 7) by injecting DT directly to the uterine lumen (similar to Science 2011 18;331(6019):912-6. 21330545) to avoid side effects on other organs.

- Thank you for this suggestion. Unfortunately, in contrast to the paper referenced, where the study was performed in adult mice, we were unable to perform DT uterine injection due to technical reasons, as the uterine tissues at P7 and P14 are much too thin even for insertion of the finest syringe available.

11. Do these GFP^{low+} cells develop organoids (Fig. 6a)? The authors should characterize endometrial organoids derived from *Lgr5*⁺ cells. Expression of secretory, ciliated, and glandular markers? Why these organoids show no glandular budding similar to intestinal organoids? *Lgr5*⁺ cells should be cultured in the presence of Wnt4 and Wnt7a/b, the Wnt ligands expressed by the uterus, and not Wnt3a, not present in the uterus. This might explain the lack of glandular budding in these organoids. The organoids cultured in the low Wnt conditions are the vacuolated type, not budded type. Similar to the intestine, glandular buddings should be attached to the luminal epithelium.

- It was not possible to distinguish GFP^{med} and GFP^{low} uterus populations by FACS. We therefore sorted the *Lgr5*-GFP^{med/low} population, presumed to represent *Lgr5*^{med/low} cells from P14 *Lgr5-2A-EGFP* mouse uterus and evaluated their ability to initiate organoid formation *in vitro*. This revealed that GFP^{med/low} cells retain some ability to generate organoids, albeit at a reduced frequency to GFP^{high} cells (4.6% vs 7.7%, P=0.01, n=5) (Figure for revision (b, c)). Since the GFP^{med/low} cells are presumptive intermediate progeny of the majority *Lgr5*^{high} gland progenitor cells, this supports a gradual loss of stemness during commitment to more differentiated gland lineages, as has been shown for other tissues such as the small intestine.

- In response to reviewer questions, we have further characterized the organoids in greater phenotypic and molecular detail when cultured under various Wnt conditions. Under normal Wnt conditions, organoids are spherical (referred as round-type in text), highly proliferative and express low levels of differentiation markers FoxA2 and Lif (Fig. 7m-o). In contrast, when cultured in low Wnt conditions, the organoids adopt an irregular shape consistent with the onset of cellular differentiation (as seen in other organoid systems, including intestine and stomach), they display limited proliferation and robust expression of FoxA2 and Lif (Fig. 7m-o). We now refer to this differentiated type of organoids as vacuolated-type as suggested by the reviewer.
- More detailed characterization of these two organoid subtypes revealed lack of luminal lineage— The early differentiation marker Prss28 was expressed at low, but comparable levels in the two organoid sub-types as compared to adult tissue, likely reflecting partial commitment to some functional glandular lineages under both culture conditions. While expression of luminal markers Wnt7a, Scnn1a and Irg1 was detectable in both culture conditions (Fig. 7m), their levels were far below the physiologically relevant levels present in adult tissue (Supplementary Fig. 7e), likely underscoring the specificity of the culture conditions for sustaining glandular progenitor cell proliferation/differentiation *ex vivo*. As growth factor requirements for luminal differentiation are not established, the lack of luminal differentiation could reflect limitations of our culture system, and could underlie the reason for the absence of budding in our organoid culture.
- While we did not observe budding organoids, Boretto et al have documented highly proliferative budding uterine organoids (Boretto, M. et al. Development 2017). There could be 2 main reasons for the differences in our organoid system and Boretto et al's – (1) Boretto et al derived organoids from adult uterus, while we generated organoids from P14 uterus, thus different organoid morphologies could be due to the different intrinsic behaviors in uterine epithelia from these two stages; (2) Boretto et al used conditioned media for Wnt and Rspo supplementation while we used exogenous factors, making it difficult to compare the relative concentrations of these growth factors in the culture conditions. Therefore, it may be unreasonable to directly compare the observations of

the two groups. In all, based on our more detailed characterization of the rounded and vacuolated-type organoids, we conclude that the rounded-type organoids under high Wnt conditions are more representative of stem/progenitor cells while the vacuolated-type organoids occurring in low Wnt conditions reflect a more differentiated glandular system.

- Supplementing Wnt4 or Wnt7b instead of Wnt3a also selectively promoted the generation of round-type organoids (Supplementary Fig. 7f-g). They also maintained similar levels of *Lgr5* and *Fzd10* expression as well as *Axin2*, indicating robust endogenous Wnt signaling activation (Supplementary Fig. 7h). This is described in the manuscript as following.

(Page14 Line309)

Supplementing Wnt4 or Wnt7b in place of Wnt3a also activated *Axin2* and selectively promoted the generation of round-type organoids as expected from their ability to upregulate *Lgr5* and *Fzd10* expression. (Supplement Fig. 7f-h). We then knocked down *Fzd10* in organoids to evaluate the functional relevance of *Fzd10* (Supplementary Fig. 7i, j). This resulted in a major induction of apoptosis throughout the organoids, consistent with a role for *Fzd10* as an essential Wnt receptor in maintaining organoid growth *ex vivo*.

12. To prove the concept of stem cell niche, the authors should use appropriate Wnt ligands (Wnt4 and Wnt7a/b) in their culture conditions. Conditioned media from Lgr5- cells is sufficient to replace Lgr5- cells?

- As described above, supplementing the organoid medium with Wnt4 and Wnt7b resulted in mostly round-type organoids, implying that these two Wnts, supplied endogenously from Lgr5- epithelial gland cells, are indeed robustly activating canonical Wnt signaling, similar to Wnt3a (Supplementary Fig 7f-g). We were unable to efficiently grow organoids supplemented with conditioned medium from Lgr5- cells. Given that intestinal niche cells such as Paneth cells promote stem cell function through direct cell contact (Sato, T. et al. Nature 2011), we hypothesize that optimal niche functions of Lgr5- endometrial cells may similarly require direct contact with Lgr5+ gland cells to modulate stem cell behaviour *in vitro*. Alternatively, close proximity of Lgr5- niche and Lgr5+ progenitor cells may be important for establishing sufficiently high local Wnt concentrations to effect efficient Wnt pathway activation. Whilst these are interesting speculations and areas of future study, they are beyond the current scope of this study.

13. Fig. 8 g ETC159 treated uterine section is close to utero-tubal junctions where there are naturally fewer glands and the vehicle-treated section from the middle of the uterus which is full of glands. The authors should provide 3D imaging or thick section whole mount imaging for this purpose.

- We utilized 3D imaging for the ETC159 treated mouse. Data are shown in Fig. 8i.

14. In vivo, the epithelial cell-specific loss of Porcupine in mice is dispensable for glandular development but required for fertility (Biol Repro 2017 29036275). The ablation of Porcupine from both stromal and epithelial cells is also dispensable for endometrial gland development but required for their maintenance (Dev Biol 2017 27965056). Data presented in Fig. 8g is opposite to previous observations. ETC159 IP injections might have affected pituitary and ovarian functions, where Wnt plays a major role, to influence uterine gland development? OR uterine sections analyzed from different depths or locations have affected the assessment as glandular buds and extensions are clearly presented in both groups (Fig. 8g). Foxa2 is a direct target of Wnt signaling (Oncogene 2013 18;32(29):3477-82 22945641). Therefore, it is not appropriate to use Foxa2 as a proxy marker for the absence or presence of glands in Wnt suppressive conditions (Fig. 8g). Ck8 staining and 3D imaging is a more appropriate approach.

- As the reviewer suggested, we utilized 3D reconstruction to image gland development for ETC159- treated mouse (Fig. 8i) to ensure that the glandular formation is not misrepresented due to technical limitations. Using this method, we confirmed that gland formation was indeed attenuated by ETC159 treatment. As described in the Discussion (Page 19 Line 423), targeting the epithelial compartment substantially impaired gland formation. ETC159 selectively inhibits Wnt signaling in the epithelial compartment *in vivo* (Madan, B. et al. Oncogene. 2016 and private communication). In fact, Farah et al proposed a dichotomy in which Wnts from stromal cells antagonize glandular development while Wnts from glandular epithelium promote glandular development. Therefore, they hypothesize that the *Porcn* mutant mouse exhibited normal gland development because of the concomitant ablation of both stromal “opposing” Wnts and epithelial “promoting” Wnts in the uterus. In light of these data, the attenuation of gland formation due to selective inhibition of epithelial Wnts by ETC159 is line with the find in Farah et al – that epithelial Wnts are important for promoting gland development and formation.

15. How Lgr5+ cells transcriptional profile differs from Lgr5- cells?

- We utilized microarray analysis for more detailed characterization (data shown in Fig.

6b). As a result, *Aldh1a1* and *Prom1* were found to be upregulated in *Lgr5*⁺ cells, and *Spr1a* and *Zfp750* were upregulated in *Lgr5*⁻ cells (Fig. 6c). *Aldh1a1* has previously been reported as a marker of glandular stem/progenitors based on single RNA sequence data (Wu, B. et al. Stem Cell Reports 2017). *Prom1* is well characterized as a stem cell marker in other organs (Snippert, HJ. et al. Gastroenterology 2009). We validated these results by qPCR and RNA ISH (Fig. 6e-g, Supplementary Fig. 6c-f). Taken together, these results further support a stem cell identity for the sorted *Lgr5*⁺ uterine gland cells and is discussed in the manuscript:

(Page 18 Line 403)

Given that *Aldh1a1* has been reported as a stem/progenitor marker in developing endometrium based on single RNA sequence analysis⁵ and also *Prom1* is an established stem cell marker in small intestine, the upregulation of these genes in *Lgr5*⁺ cells further supports their stem cell identity³⁷.

16. Data presented on the vagina is very distracting, given not much has been done on this organ. This data could be presented as supplementary data.

- Thank you for the constructive comment. We have moved this data to supplementary figures (Supplementary Fig. 1c).

17. Current norm is at least 5 mice per time point compared to 3.

- Thank you for valuable comment. We have increased the n-number to 5 for all major experimental data.

Reviewer #3 (Remarks to the Author):

The manuscript by Seishima and coworkers identifies the *Lgr5*⁺ cell in the developing uterus and uterine glands as stem/progenitor cell for the mouse uterus. This is a very significant and important finding. The work is thorough. The group used lineage marking, organoids and cell ablation as a means of demonstrating these cells are uterine glandular stem cells. This group also shows that it is paracrine WNT signaling that regulates the proliferation of these stem cells. The only issue that would be nice for this group to resolve is the co localization of these cells with Estrogen and Progesterone receptor since both regulate gland development. Other than one small issue this is an excellent manuscript.

- We performed co-staining for EGFP and Estrogen and Progesterone receptors on P14

Lgr5-2A-EGFP mouse uterus (Figure for revision g). ER expression was mostly restricted to the stroma and only a small number of glandular cells were positive for both ER and EGFP. It will be interesting to further explore any specific function of this minor ER+EGFP+ population in future studies. In contrast, PgR was broadly expressed throughout the epithelium, and accordingly displayed a partial overlap with Lgr5-EGFP.

Reviewers' Comments:

Reviewer #1:

Remarks to the Author:

OVERALL AND MAJOR COMMENTS

This revised manuscript is much improved and has sufficiently answered most of the comments from the initial review. The exception is that the Materials and Methods are not sufficient to allow for another laboratory to reproduce the experiments in a rigorous manner.

Major Comments:

(1) Figure 3 and elsewhere: The precise cell types in the uterine wall need to be labeled for the reader including luminal epithelium (le), glandular epithelium (ge), stroma and/or myometrium where applicable.

(2) Figure 7: The immunostaining for Foxa2 and Lif in Panel n is not convincing.

(3) The Methods need to be carefully revised to ensure rigor and reproducibility. Exact information on all chemicals and reagents including source and catalog numbers needs to be provided. In example:

a. Line 480: What is the location of Prof. D.M. Virship? Was the 50% PEG400 in water?

b. Line 490: What type of collagenase? What was the source?

c. The microarray data needs to be deposited in a publicly accessible repository.

d. Line 573: What type of pressure cooker?

e. Line 600: What is the source of the RapiClear?

SPECIFIC COMMENTS

Line Comment

39 A two sentence paragraph seems a bit short

44 Calcitonin has not been shown to be important using a genetic knockout strategy. Recently, a number of papers published by Kelleher and coworkers have demonstrated a role of uterine glands in pregnancy establishment. Those should be cited in the Introduction and Discussion.

Suppl. Fig. 7a Promoter is not spelled correctly

Reviewer #2:

Remarks to the Author:

This revised manuscript by Seishima et al., has addressed some key concerns and is significantly improved. In particular, the 3D imaging and organoid culture analysis is more refined and the unsupported conclusions regarding stem/progenitor potential of Lgr5+ cells are now better defined. However, as detailed below, multiple points raised from the first submission have not been satisfactorily addressed and key observations have not been investigated in sufficient depth. As such, additional data is still required to support the authors conclusions and provided model. There are two main claims in this paper (Lgr5+ stem cells and the concept of niche), which are currently unsupported by the presented data.

Issues with the current interpretation of Lgr5+ as stem/progenitor cells

1. As highlighted in my previous comments, multiple studies using cell lineage tracing approach have already proven that Mullerian duct epithelium gives rise to both glandular and luminal epithelium of the uterus. For example, cell lineage tracing of Wnt4+ cells showed that Wnt4EGFPCre+ progenitor daughter cell give rise to both luminal and glandular epithelium. This study using another Wnt-related gene, Lgr5, is essential showing the same (Fig. 1 and 2 are

similar to Wnt4 data in Hum Mol Genet. 2016 Mar 15;25(6):1059-73 26721931). The key advancement would have been the demonstration if there are a subset of cells that express Lgr5+ and these are cells that act as stem/progenitor cells. Otherwise, this study is a repetition of many other studies that have extensively proven that Mullerian duct epithelium gives rise to all the epithelial cells of the uterus.

2. In the intestine, Shyer et al (Cell 2015) showed that Lgr5 is present in all the epithelial cells during embryonic stages and its expression gets restricted after villi formation. In contrast, Nigmatullina et al Embo J 2017 showed that Lgr5 expression was restricted to some but not all the cells of the intestine leading to a major confusion in the field regarding the validity of Shyer et al paper. Therefore, the authors should perform detailed analysis and quantification of Lgr5+ cells in the Mullerian duct epithelium and decipher the heterogeneity of this epithelium.

3. Abstract presented at the International Society for Stem Cell Research (ISSCR) annual meeting 2018 by a well-established group in endometrial stem cells showed that Lgr5+ cells are not stem cells in the adult uterus. Although this study is mainly concerned with the neonatal uterus, it is essential a more thorough investigation is performed of Lgr5 expression and Mullerian duct epithelial heterogeneity before claiming that these cells are stem/progenitor cells.

4. The ability to replace lost tissue through cell division is a major characteristic of stem cells (Clevers Cell Stem Cell 2019). The authors have provided no evidence to show that these Lgr5+ cells persist after injury? Proliferate? Respond to hormones? PR is not a marker of the luminal epithelium (Fig. 5i) and the data presented in revision d-f is misleading as the authors have provided no evidence that glandular epithelium are depleted.

5. 10. The author should provide in-vivo data from Lgr5-2A-DTR ablation model (Fig. 7) by injecting DT directly to the uterine lumen (similar to Science 2011 18;331(6019):912-6. 21330545) to avoid side effects on other organs.

- Thank you for this suggestion. Unfortunately, in contrast to the paper referenced, where the study was performed in adult mice, we were unable to perform DT uterine injection due to technical reasons, as the uterine tissues at P7 and P14 are much too thin even for insertion of the finest syringe available.

This is incorrect. Many groups regularly (Saatcioglu et al Elife 2019) perform similar surgeries even in neonatal mice/rats.

Issues with the concept of stem cell niche

1. The authors need to provide data showing that Wnt7b and Wnt4 supplemented organoids are similar to Wnt3a supplemented organoids. Wnt signaling is not required for growth but maintenance of endometrial organoids (Boretto Development 2017). Therefore, passaging data and marker analysis of organoids is essential to support their claim. It is expected that these Wnt ligands will stimulate Wnt target genes in a comparable manner (SFig. 7). Additionally, SFig. 7j don't look like organoids.

2. In 3D (Fig. 8i), ETC159 treatment is causing ablation of both luminal and glandular epithelium. In contrast, histological sections (Fig. 8g) is only showing the absence of glands. Does this suggest that ETC159 treatment causes the loss of both luminal and glandular epithelium??

3. "We were unable to efficiently grow organoids supplemented with conditioned medium from Lgr5- cells". This suggests that Lgr5- cells are unlikely to act as niche cells. If these cells are needed to be in close proximity, then cell culture inserts can be utilized to address this issue.

Minor concern

1. The author should quantify the Ki67 data between tamoxifen-treated and untreated mice to show that there is no difference between uteri from these two groups. This data should be presented in the paper as this issue will be of concern for readers. The picture presented are showing a major increase in Ki67+ cells in tamoxifen injected mice compared to controls.

Reviewer #3:

Remarks to the Author:

This is an exciting manuscript that has addressed as well as possible the comments of the initial review. It will be of broad interest to the Stem cell community as well as those studying female reproduction.

We thank the reviewers for their compliments on the improvement in manuscript quality and constructive comments. We agree with the Editor that the injury experiment is not necessary for the manuscript, and, as detailed below, we do not think that additional experiments suggested by Reviewer 2 will add further value and have accordingly provided detailed reasons to justify our perspective for each point. We have addressed all the other comments, highlighted the changes in the revised manuscript and referenced the manuscript/figure changes in the corresponding sections in the rebuttal letter. We hope that the Editor will be convinced by our reasoning and revision, and now deem it suitable for publication in Nature Communications.

Neonatal Wnt-dependent LGR5+ Stem/Progenitor Cells, Supported by Epithelial Niche Cells, are Essential for Uterine Gland Development

Reviewer #1 (Remarks to the Author):

We have addressed all of Reviewer 1's comments accordingly in our revised version.

OVERALL AND MAJOR COMMENTS

This revised manuscript is much improved and has sufficiently answered most of the comments from the initial review. The exception is that the Materials and Methods are not sufficient to allow for another laboratory to reproduce the experiments in a rigorous manner.

Major Comments:

(1) Figure 3 and elsewhere: The precise cell types in the uterine wall need to be labeled for the reader including luminal epithelium (le), glandular epithelium (ge), stroma and/or myometrium where applicable.

Thank you for the suggestion. We labeled as recommended to make it easy for the readers to understand. (Fig. 3d,f,h,j, Fig. 4b-e, Fig. 6e,h, Supplementary Fig. 2a-d) As additional lines would obscure the RNA ISH and IHC signals, we tried to describe the anatomical locations to the reader by better annotating the schematic (Fig. 3l) for the reader's reference. For Fig. 4, we are happy to add another schematic to annotate the regions if necessary.

(2) Figure 7: The immunostaining for Foxa2 and Lif in Panel n is not convincing.

We have replaced the previous images with ones with stronger staining. (Fig. 7n,o)

(3) The Methods need to be carefully revised to ensure rigor and reproducibility. Exact information on

all chemicals and reagents including source and catalog numbers needs to be provided. In example:
Thank you for pointing this out. We have revised our Materials and Methods accordingly to include the requested detailed information.

a. Line 480: What is the location of Prof. D.M. Virship? Was the 50% PEG400 in water?

Line 483

ETC159 (gift from Prof. D.M. Virshup, Duke NUS Medical School) was dissolved in 50% PEG400 (v/v) in water and was injected s.c. daily at a dose of 10 mg/kg body weight.

b. Line 490: What type of collagenase? What was the source?

Line 494

2 mg/ml collagenase I (Worthington)

c. The microarray data needs to be deposited in a publicly accessible repository.

Our apologies for this oversight. We have submitted the data to GEO and received the accession number (GSE137974). We added to the manuscript.

Line 631

Data availability. Microarray data that support the findings of this study have been deposited in the Gene Expression Omnibus (GEO) under accession code GSE137974.

d. Line 573: What type of pressure cooker?

Line 577

Antigen retrieval was carried out by heating slides at 121 °C for 20 min using 2100 Antigen Retriever (Aptum Biologics)

e. Line 600: What is the source of the RapiClear?

Line 605

Tissues were then cleared with RapiClear 1.52 (Sunjin Lab)

SPECIFIC COMMENTS

Line Comment

39 A two sentence paragraph seems a bit short

44 Calcitonin has not been shown to be important using a genetic knockout strategy. Recently, a number of papers published by Kelleher and coworkers have demonstrated a role of uterine glands in pregnancy establishment. Those should be cited in the Introduction and Discussion.

Thank you very much for the suggestion. We have revised the Introduction as follows:

Line 39-

The majority of the female reproductive tract, comprising the oviduct, uterus, cervix and upper vagina, develops from the Müllerian duct (Md) during embryogenesis¹. These organs remain relatively immature at birth and undergo further development during prepuberty to ensure fertility in adulthood. Particularly in uterus, gland development is essential for successful pregnancy as their secretions and products impact on implantation, stromal cell decidualization and placental development².

Uterine glands are essential for proper uterine function as they secrete various factors such as Leukemia Inhibitory Factor (LIF) and Fibroblast growth factor (FGF) that are important for endometrial receptivity and embryo implantation^{3,4}. Their development begins postnatally, involving budding, tubulogenesis, coiling and branching of luminal epithelia, orchestrated by interactions with the underlying stroma⁵. Genetic knockout studies have identified multiple, predominantly Wnt-related, genes required for development of the uterine epithelium⁶, but the cellular origins of the glandular epithelium remain poorly understood. Although single cell sequencing studies have recently documented extensive cellular heterogeneity within the developing mouse uterus epithelium, it is currently unknown whether dedicated endometrial stem/progenitor cells are present⁷.

1. Mullen, R.D. & Behringer, R.R. Molecular genetics of Müllerian duct formation, regression and differentiation. *Sex Dev.* **8**, 281–296 (2014).
2. Kelleher, A.M., DeMayo, F.J., Spencer, T.E. Uterine glands: Developmental biology and functional roles in pregnancy. *Endocr Rev.* doi: 10.1210/er.2018-00281. [Epub ahead of print] (2019).
3. Stewart, C.L. et al. Blastocyst implantation depends on maternal expression of leukaemia inhibitory factor. *Nature* **359**, 76-79 (1992).
4. Roberts, R.M. & Fisher, S.J. Trophoblast stem cells. *Biol. Reprod.* **84**, 412–421 (2011).
5. Gray, C.A. et al. Developmental biology of uterine glands. *Biol. Reprod.* **65**, 1311–1323 (2001).
6. Kobayashi, A. & Behringer, R.R. Developmental genetics of the female reproductive tract in mammals. *Nat. Rev. Genet.* **4**, 969–980 (2003).
7. Wu, B. et al. Reconstructing Lineage Hierarchies of Mouse Uterus Epithelial Development Using Single-Cell Analysis. *Stem Cell Reports* **9**, 381–396 (2017).

Suppl. Fig. 7a Promoter is not spelled correctly

We corrected the spelling accordingly. (Supplementary Fig. 7a)

Reviewer #2 (Remarks to the Author):

This revised manuscript by Seishima et al., has addressed some key concerns and is significantly improved. In particular, the 3D imaging and organoid culture analysis is more refined and the unsupported conclusions regarding stem/progenitor potential of Lgr5⁺ cells are now better defined. However, as detailed below, multiple points raised from the first submission have not been satisfactorily addressed and key observations have not been investigated in sufficient depth. As such, additional data is still required to support the authors conclusions and provided model. There are two main claims in this paper (Lgr5⁺ stem cells and the concept of niche), which are currently unsupported by the presented data.

While we are happy that Reviewer 2 has acknowledged the improvements in our manuscript, we respectfully, but robustly disagree with the majority of his/her comments and accordingly provide detailed explanation and support for our stance in a point-by-point format as follows. In summary, the key novel findings in this manuscript are (i) Lgr5⁺ cells in the *neonatal* uterus are indispensable stem cells for uterus development and (ii) these Lgr5⁺ stem cells are supported by an epithelial niche. Therefore, while we have included data from Mullerian duct and adult uterus to complete the characterization of Lgr5⁺ cells across various developmental stages, these are not the focus of the manuscript and we do not think that it is necessary or fruitful to pursue these characterizations in further detail in this manuscript. Strikingly, the fact that all three reviewers have no more questions concerning the role of Lgr5⁺ stem cells in the neonatal uterus indicates that we have convinced them of the major finding of our manuscript.

Issues with the current interpretation of Lgr5⁺ as stem/progenitor cells

1. As highlighted in my previous comments, multiple studies using cell lineage tracing approach have already proven that Mullerian duct epithelium gives rise to both glandular and luminal epithelium of the uterus. For example, cell lineage tracing of Wnt4⁺ cells showed that Wnt4EGFPCre⁺ progenitor daughter cell give rise to both luminal and glandular epithelium. This study using another Wnt-related gene, Lgr5, is essential showing the same (Fig. 1 and 2 are similar to Wnt4 data in Hum Mol Genet. 2016 Mar 15;25(6):1059-73 26721931). The key advancement would have been the demonstration if there are a subset of cells that express Lgr5⁺ and these are cells act as stem/progenitor cells. Otherwise, this study is a repetition of many other studies that have extensively proven that Mullerian duct epithelium gives rise to all the epithelial cells of the uterus.

As mentioned earlier, our study started out with a holistic characterization of Lgr5 expression throughout the various stages of uterus development. This revealed distinct expression patterns in the

neonatal uterus (main focus of our study), embryonic MD and adult uterus. In the MD, we found that Lgr5 is uniformly expressed throughout the whole tissue, and we acknowledge that the tracing results from these Lgr5⁺ cells are similar to those using Wnt4a marker, which further confirms that MD cells act as stem cells. It would be erroneous, however, to assume that Wnt signaling is synonymous with Lgr5 expression, as there are many instances such as in pancreatic and cardiovascular organogenesis where Wnt signaling plays an active role without the involvement of Lgr5. Therefore, we strongly disagree with the reviewer that this significant finding is merely a “repetition” of other studies. As discussed in the earlier rebuttal, phenotypic heterogeneity within a given population, including resident stem cells, is not an accurate indicator of function. All endogenous stem cell populations present phenotypic heterogeneity due to multilineage priming and cell intrinsic/niche-dictated proliferation/division modes. Accordingly, we do not believe that analysis of phenotypic heterogeneity (if any) within the Mullerian duct Lgr5⁺ population via single cell RNAseq would be informative in terms of further refining stem cell identity. Finally, the MD constitutes a very minor part of the paper and does not justify the expensive, long-term investment required of the experiments suggested by the reviewer.

2. In the intestine, Shyer et al (Cell 2015) showed that Lgr5 is present in all the epithelial cells during embryonic stages and its expression get restricted after villi formation. In contrast, Nigmatullina et al Embo J 2017 showed that Lgr5 expression was restricted to some but not all the cells of the intestine leading to a major confusion in the field regarding the validity of Shyer et al paper. Therefore, the authors should perform detailed analysis and quantification of Lgr5⁺ cells in the Mullerian duct epithelium and decipher the heterogeneity of this epithelium.

The two papers cited by the reviewer were performed using a previous Lgr5-eGFP-IresCreERT2 mouse model with known transgene silencing and variegated expression. In contrast, the Lgr52ACreERT2 model used here accurately reports all endogenous Lgr5 expression (Leushacke et al., 2017) and consequently leaves no room for ambiguity. Due to reviewer concerns, we have replaced the previous image Fig1d-e with one that is clearer, showing uniform Lgr5-eGFP expression throughout the Md. We therefore do not think that further analysis or quantitation of MD cells will enhance our results or yield any significant finding. Again, we reiterate that the MD is not the main focus of this study and we therefore deem it beyond the scope of this paper to investigate possible heterogeneity (if any) in the Lgr5⁺ population in the MD.

3. Abstract presented at the International Society for Stem Cell Research (ISSCR) annual meeting 2018 by a well-established group in endometrial stem cells showed that Lgr5⁺ cells are not stem cells in the adult uterus. Although this study is mainly concerned with the neonatal uterus, it is essential a more thorough investigation is performed of Lgr5 expression and Mullerian duct epithelial

heterogeneity before claiming that these cells are stem/progenitor cells.

Our findings presented in Fig S4d agree with the (unpublished) ISSCR abstract that adult Lgr5⁺ cells in the uterus are not stem cells under homeostatic conditions. Through lineage tracing, we demonstrated that adult Lgr5⁺ cells are not maintained in the uterus and we therefore do not claim that the Lgr5⁺ cells act as stem cells in the adult uterus. However, as different developmental stages have different dynamics, tissue requirements and turnover rates, this observation in the adult uterus does not preclude Lgr5⁺ cells from being stem/progenitor cells at other developmental stages, such as the neonatal period.

As mentioned in the two previous points, we observed uniform Lgr5 expression throughout the MD; not only is further analysis of MD cells for a hypothesized heterogeneity beyond the scope of our study, it does not add value to our main finding that neonatal Lgr5⁺ cells contribute to glandular and luminal epithelium in a stage-dependent manner or the manuscript as a whole.

For clarity, we have replaced Fig. 1d-e with a more representative image showing uniform Lgr5-eGFP expression in the Md.

4. The ability to replace lost tissue through cell division is a major characteristic of stem cells (Cell Stem Cell 2019). The authors have provided no evidence to show that these Lgr5⁺ cells persist after injury? Proliferate? Respond to hormones? PR is not a marker of the luminal epithelium (Fig. 5i) and the data presented in revision d-f is misleading as the authors have provided no evidence that glandular epithelial are depleted.

While the recent Cell Stem Cell review does define stem cells as cells with ability to replace lost tissue, this “loss” can be induced by injury, or the regular renewal of the tissue. While the former represents an acute loss of tissue that requires a robust and rapid response by facultative stem cells which do not exhibit stem cell behavior under normal conditions, such as Lgr5⁺ chief cells in the stomach corpus (Leushacke et al., 2017), the function of homeostatic stem cells in regular tissue renewal is only observable after multiple rounds of tissue renewal (e.g. Lgr5⁺ cells in the small intestine, stomach pylorus, hair follicles). We have performed lineage tracing from the neonatal uterus from P7 (and P14) to P56, spanning multiple rounds of tissue renewal into adulthood (an estrous cycle is 4-5 days in rodents), indicating that Lgr5⁺ cells in the neonatal uterus are stem cells. Given the different modes of action of facultative and homeostatic stem cells, we agree with the Editor that it is not necessary to utilize an injury model in our study as our focus is on homeostatic neonatal stem cells.

The PR staining was performed at the suggestion of Reviewer 3. We intended to use PR expression to show that luminal epithelium is unaffected by Lgr5⁺ cell ablation not only histologically but also functionally; we had not intended for the use of PR as a luminal epithelial marker. We will remove

these images if they are confusing for the readers.

5. 10. The author should provide in-vivo data from Lgr5-2A-DTR ablation model (Fig. 7) by injecting DT directly to the uterine lumen (similar to Science 2011 18;331(6019):912-6. 21330545) to avoid side effects on other organs.

- Thank you for this suggestion. Unfortunately, in contrast to the paper referenced, where the study was performed in adult mice, we were unable to perform DT uterine injection due to technical reasons, as the uterine tissues at P7 and P14 are much too thin even for insertion of the finest syringe available.

This is incorrect. Many groups regularly (Saatcioglu et al Elife 2019) perform similar surgeries even in neonatal mice/rats.

We emphasize once again that there is no study that describes successful intra-uterine injections in neonatal pups, including the two studies cited by the Reviewer. The Science paper cited (Science 2011 18;331(6019):912-6. 21330545) performed *in utero* injections on pregnant adult female mice, while the Elife paper cited here (Saatcioglu et al Elife 2019) described subcutaneous injections into rat pups. Therefore, there is no precedent to demonstrate that it is technically feasible to inject directly into the uterine lumen of neonatal mouse pups.

Issues with the concept of stem cell niche

1. The authors need to provide data showing that Wnt7b and Wnt4 supplemented organoids are similar to Wnt3a supplemented organoids. Wnt signaling is not required for growth but maintenance of endometrial organoids (Boretto Development 2017). Therefore, passaging data and marker analysis of organoids is essential to support their claim. It is expected that these Wnt ligands will stimulate Wnt target genes in a comparable manner (SFig. 7). Additionally, SFig. 7j don't look like organoids.

In our revised manuscript, we showed that culturing the organoids with Wnt4/7b has a similar effect as Wnt3a in terms of inducing Wnt signaling activity, as demonstrated by expression levels of the Wnt target genes Lgr5 and Axin2 (Sup Fig. 7). Wnt4/7b-supplemented organoids can also be passaged for at least 5 passages over 2 months, similar to Wnt3a-supplemented organoids. Based on these observations, we are very confident in claiming that Wnt4/7b is functionally equivalent to Wnt3a in supporting the organoids.

The Reviewer's comment questioning whether Fig S7j shows organoids is puzzling and frankly somewhat insulting. These are indeed 3D organoids, which have been flattened under a coverslip in order to visualize the immunostaining.

2. In 3D (Fig. 8i), ETC159 treatment is causing ablation of both luminal and glandular epithelium. In contrast, histological sections (Fig. 8g) is only showing the absence of glands. Does this suggest that ETC159 treatment causes the loss of both luminal and glandular epithelium??

The 3D image in Fig. 8i uses K8 staining (as suggested by the reviewer) to highlight the abnormalities in the glandular epithelium seen after ETC treatment. The insert in the control image shows normal glands, whereas the image with ETC159-treated uterus has lost glandular structure. The luminal epithelium is not affected by ETC159 treatment. We have now replaced the previous image with a better-rendered image showing an intact luminal epithelium in ETC159-treated uterus. (Fig. 8i) We also added a comment as followed;

Line 863

In contrast, LE development was largely unaffected by ETC159 treatment.

3. “We were unable to efficiently grow organoids supplemented with conditioned medium from Lgr5- cells” . This suggest that Lgr5- cells are unlikely to act as niche cells. If these cells are needed to be in close proximity, then cell culture inserts can be utilized to address this issue.

The statement above was our response in the previous rebuttal letter after we attempted the use of conditioned media from Lgr5- cells suggested by the reviewer. We robustly disagree that this single experiment leads to the conclusion that Lgr5- cells are unlikely to be niche cells. There could be many reasons why this was not technically optimal – for example, the culture conditions for Lgr5- cells have not been optimized for their growth and/or maximal Wnt secretion/stability. Hence, it is perfectly plausible that conditioned media generated from Lgr5- cultures simply do not contain a high enough active Wnt concentration to support uterine organoid growth; or the Lgr5- niche cell needs to be in close/direct contact with the Lgr5+ cells such as in the case of Paneth cell and small intestinal stem cell (which cannot be addressed with cell culture inserts).

In contrast, we have a large, robust body of data to prove that the Wnt4/7b secreted by Lgr5- epithelial cells support Lgr5+ stem cells –

- (i) Functional data that demonstrate Wnt4/7b are able to support uterine organoid culture (Sup Fig 7f,g)
- (ii) Expression of Wnt 4/7b by Lgr5- cells (Fig 6f,g)
- (iii) Lgr5+ cells co-cultured with Lgr5- epithelial cells in low Wnt conditions generate organoids more efficiently than Lgr5+ cells cultured alone (Fig 8c)
- (iv) Lgr5+ cells co-cultured with Lgr5- epithelial cells in low Wnt conditions generate larger

and more rounded organoids than Lgr5⁺ cells cultured alone (Fig 8d,e).

- (v) Addition of ETC159 (which inhibits Wnt secretion) to the co-culture conditions caused a decrease in organoid outgrowth efficacy, average organoid size, and more vacuolated organoids (Fig 8c-e).

Collectively, this data strongly supports Wnt4/7b secreted by Lgr5⁻ epithelial cells as being an important endogenous Wnt niche source for Lgr5⁺ stem cells, which simply cannot be refuted based on a single, technically suboptimal conditioned media experiment.

Minor concern

1. The author should quantify the Ki67 data between tamoxifen-treated and untreated mice to show that there is no difference between uteri from these two groups. This data should be presented in the paper as this issue will be of concern for readers. The picture presented are showing a major increase in Ki67⁺ cells in tamoxifen injected mice compared to controls.

We have performed quantification of Ki67 cells in the revised manuscript (Supplementary Fig. 4d,e). This conclusively shows no difference in proliferation index between 4OHT-injected and untreated mice in both the epithelial and stromal cell populations.

Reviewers' Comments:

Reviewer #1:

Remarks to the Author:

OVERALL COMMENTS

The authors have satisfactorily addressed most comments from the initial review. A major lingering issue is that lack of enough detail in the Materials and Methods to allow for another lab to reproduce the findings using the same protocols and reagents in a rigorous manner.

MAJOR COMMENTS

(1) The paper should discuss the findings of Shiyong Jin "Bipotent stem cells support the cyclical regeneration of endometrial epithelium of the murine uterus" published in Proceedings of the National Academy of Sciences 2018.

(2) Figure 8g: The Foxa2 IHC displayed in the vehicle panel is of low quality compared to many of the other figures such as 5h. Perhaps this data should be moved to the Supplement.

(3) Materials and Methods: Catalog numbers for indicated reagents should be provided for the reader. For example, which specific collagenase I from Worthington? Detailed information for all antibodies and reagents should be provided to allow another laboratory to reproduce the findings in a rigorous manner.

SPECIFIC COMMENTS

Line Comment

46 Remove FGFs, as they have not been shown to be critical for implantation. Only LIF has conclusively been demonstrated to be an essential glandular-derived factor in mice.

483 The entire name of Dr. Vishup should be provided as well as exact location in the world and contact information so others can obtain this compound

Our point-by-point response to the reviewer is as follows:

Reviewer #1 (Remarks to the Author):

OVERALL COMMENTS

The authors have satisfactorily addressed most comments from the initial review. A major lingering issue is that lack of enough detail in the Materials and Methods to allow for another lab to reproduce the findings using the same protocols and reagents in a rigorous manner.

Major Comments:

(1) The paper should discuss the findings of Shiyong Jin “Bipotent stem cells support the cyclical regeneration of endometrial epithelium of the murine uterus” published in Proceedings of the National Academy of Sciences 2018.

Thank you for the suggestion. We made a comment in the discussion as followed;

Line 395-

It was recently reported that bipotent stem cells in adult endometrium reside in the zone between GE and LE, but further investigation will be needed to better characterize them³⁷. Given that the Lgr5⁺ cells are found in the developing GE residing proximally to Foxa2+ GE cells, it would be interesting to investigate whether the Lgr5⁺ postnatal progenitors can be the source of bipotent stem cells in the adult.

(2) Figure 8g: The Foxa2 IHC displayed in the vehicle panel is of low quality compared to many of the other figures such as 5h. Perhaps this data should be moved to the Supplement.

As this image is important for showing the major difference between the two treatments, we think that it needs to be in the main figure. To better clarify for the reader, we have highlighted the FOXA2 expression as inserts. We hope this suffices.

(3) Materials and Methods: Catalog numbers for indicated reagents should be provided for the reader. For example, which specific collagenase I from Worthington? Detailed information for all antibodies and reagents should be provided to allow another laboratory to reproduce the findings in a rigorous manner.

Thank you for the suggestion. We have added the catalog number information to the reagents.

SPECIFIC COMMENTS

Line Comment

46 Remove FGFs, as they have not been shown to be critical for implantation. Only Lif has conclusively been demonstrated to be an essential glandular-derived factor in mice.

Thank you for the suggestion. We have removed the description about FGF.

Line 45 now reads -

Uterine glands are essential for proper uterine function as they secrete various factors such as Leukemia Inhibitory Factor (LIF) that are important for endometrial receptivity and embryo implantation³.

483 The entire name of Dr. Vishup should be provided as well as exact location in the world and contact information so others can obtain this compound

Thank you for the suggestion. We have added more information about Dr. Virshup.

Line 478 - ETC159 (gift from Prof. David.M. Virshup, Duke NUS Medical School, Singapore)